# MRPO: Magnitude-Regularized Policy Optimization via L1 Constraints

Wei Han [1]   Yuanxing Liu [1]   Mingda Li [1]   Ruiyu Xiao [1]   Weinan Zhang [1 2]   Ting Liu [1]

## Abstract

Reinforcement learning (RL) for large language models (LLMs) relies on imperfect reward supervision, necessitating constraints on policy updates to prevent overfitting. Nevertheless, the widely adopted KL constraint over-penalizes actions with low reference probabilities and lacks the sparsity to discard marginal policy shifts. In contrast, the L1-norm offers a distinct mechanism that is more tolerant of low-probability actions yet strictly suppresses minor probability perturbations. Motivated by this, we propose **M**agnitude-**R**egularized **P**olicy **O**ptimization (MRPO), which enforces an L1-norm constraint on policy updates. We demonstrate that MRPO permits substantial probability boosts for low-probability actions and induces sparse updates, ensuring invariance to noise that preserves the top-ranking order. Furthermore, MRPO admits a TRPO-style monotonic improvement bound under standard regularity assumptions and achieves a tighter approach to optimality than KL-based methods in single-step scenarios. Empirically, MRPO delivers exceptional results across diverse scenarios, notably doubling the performance gains of GRPO in preference alignment, outperforming DAPO in mathematical reasoning, and surpassing DPO in offline settings using only binary rewards. Code is available at https://github.com/AragornHorse/MRPO.

## 1. Introduction

Reinforcement learning (RL) has emerged as a critical post-training paradigm for large language models (LLMs) (Dubey et al., 2024; Team et al., 2025; Bai et al., 2023; Jiang et al., 2023; Comanici et al., 2025; Zhao et al., 2026; Qin et al., 2026), which aligns models with human

[1]Research Center for Social Computing and Interactive Robotics, Harbin Institute of Technology, Harbin, China [2]Suzhou Research Institute, Harbin Institute of Technology, Suzhou, China. Correspondence to: Weinan Zhang <wnzhang@ir.hit.edu.cn>.

*Proceedings of the 43rd International Conference on Machine Learning*, Seoul, South Korea. PMLR 306, 2026. Copyright 2026 by the author(s).

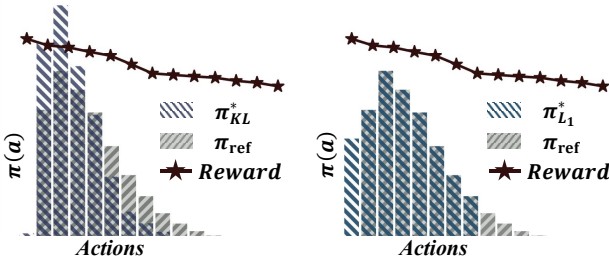

*Figure 1.* Comparison of the optimal policies under KL ($\pi^*_{KL}$) and L1 ($\pi^*_{L_1}$) constraints, with actions sorted by reward. In contrast to $\pi^*_{KL}$, $\pi^*_{L_1}$ allows for substantially larger probability boosts for high-reward actions with low reference probability (the leftmost action), while strictly maintaining the reference probabilities for a wider range (intermediate actions).

preferences (Chaudhari et al., 2025; Wang & Sun, 2025; Zhu et al., 2025) and enhances reasoning capabilities (Guo et al., 2025) by maximizing expected rewards. Since reward signals provide only weak supervision, constraining the deviation from a reference policy is critical to prevent overfitting and mode collapse (Rafailov et al., 2023). Drawing inspiration from traditional deep RL algorithms (Schulman et al., 2015; 2017), existing methodologies such as DPO (Rafailov et al., 2023) and GRPO (Shao et al., 2024), almost exclusively employ the reverse KL divergence as the constraint term. However, due to the long-tailed nature of LLM policies and the inherent imperfections of reward models (Miao et al., 2024), the KL constraint introduces fundamental limitations.

Specifically, we identify two primary limitations, as illustrated in the left panel of Figure 1: **(1) Hinders promotion of low-prior optimal actions.** When the reference probability is negligible, even a slight increase in probability can cause the KL divergence to explode. Consequently, it becomes prohibitively difficult to greatly boost the probability of high-reward actions that have low priors. Given that this is a ubiquitous phenomenon in LLMs (e.g., when questions are too challenging to answer correctly), this limitation severely stifles potential policy improvements. **(2) Sensitivity to reward noise.** The updated policy under the KL constraint ($\pi^*_{KL}$) alters the probabilities of nearly all actions, with the magnitude of change determined by the absolute reward values. Consequently, even minor reward noise shifts the solution, and the aggregated impact over the

full vocabulary can lead to substantial policy distortion.

To address these limitations, we study the geometric properties of different constraint terms and identify a key source of the limitations of the KL constraint. That also motivates the use of the L1 constraint. Based on these insights, we propose **M**agnitude-**R**egularized **P**olicy **O**ptimization (MRPO), a novel RL algorithm motivated by the geometry of L1-constrained policy optimization. Crucially, the L1-norm induces substantial yet sparse policy updates (Figure 1, right), thereby mitigating the limitations of KL regularization. Firstly, probabilities of actions above or below specific reward thresholds are substantially adjusted irrespective of their initial likelihood, enabling effective policy updates even for actions with small reference probabilities. Secondly, actions with intermediate rewards strictly retain their reference probabilities, rendering the policy invariant to reward noise that fails to alter the top-ranking order.

To rigorously validate MRPO, we provide a comprehensive theoretical and empirical analysis. We begin by comparing the geometric properties of L1 and KL constraints. We find that the L1 constraint is stricter for small policy perturbations, whereas the KL constraint becomes more restrictive for large perturbations. Therefore, actions with intermediate rewards that would undergo minor updates under KL face stricter penalties under the L1-norm, whereas extreme-reward actions requiring substantial updates encounter less resistance. Next, we investigate the convergence properties of MRPO. We find that the L1-norm not only admits a TRPO-style monotonic-improvement lower bound under standard regularity assumptions (Schulman et al., 2015), but also moves closer to the optimal policy than KL-based methods in certain settings that commonly arise in LLM training. Empirically, we evaluate MRPO across three major RL scenarios for LLMs. Results demonstrate that MRPO consistently outperforms strong baselines, notably achieving nearly double the performance gains of GRPO (Shao et al., 2024) in preference alignment, surpassing strong baselines including DAPO (Yu et al., 2026) in reasoning tasks, and outperforming DPO (Rafailov et al., 2023) in offline settings using only binary rewards ($\pm 1$).

We summarize our main contributions as follows:

- We identify the limitations of the KL constraint in LLMs, theoretically clarifying why it hinders the learning of new actions and is sensitive to reward noise.

- We propose MRPO by replacing the KL constraint with the L1-norm, and theoretically prove that it effectively overcomes the identified KL limitations.

- We apply MRPO to RLHF, RLVR, and offline RL scenarios, significantly improving the post-training effectiveness of LLMs.

## 2. Related Work

The Kullback-Leibler (KL) divergence was originally utilized in traditional deep reinforcement learning (DRL) algorithms (Schulman et al., 2015; 2017) to ensure convergence by constraining the magnitude of policy updates. Influenced by DRL, most post-training RL algorithms for LLMs employ the reverse KL divergence to penalize excessive deviation from the reference policy. Prominent examples include online algorithms like PPO-RLHF (Achiam et al., 2023), GRPO (Shao et al., 2024), and GHPO (Liu et al., 2025), as well as offline approaches such as DPO (Rafailov et al., 2023), KTO (Ethayarajh et al., 2024), and BCO (Jung et al., 2025). However, the mode-seeking nature of reverse KL tends to constrict the policy search space and hinder exploration, prompting investigations into alternative constraints like the forward KL or Jensen-Shannon (JS) divergence (Li et al., 2025). Nevertheless, these divergences remain fundamentally ratio-based measures that impose inconsistent penalties across actions with varying reference probabilities, thus failing to fully alleviate the inherent drawbacks of the KL constraint. While other constraint formulations (Terpin et al., 2022; Moskovitz et al., 2020; Pacchiano et al., 2020) exist in the broader DRL literature, they typically involve complex constrained optimization techniques that are computationally impractical for LLM training. In contrast, our proposed MRPO directly employs an L1-norm penalty, offering a simple yet effective solution that directly addresses these limitations in the considered settings without introducing optimization complexity.

## 3. Preliminary

**Reinforcement Learning for LLMs.** Mainstream RL algorithms for LLMs typically formulate the problem as a one-step Markov decision process (MDP). The prompt serves as the state $s$, and the generated response constitutes the action $a$. A reward $r(s, a)$ derived from preference models (Achiam et al., 2023) or verifiers (Guo et al., 2025), evaluates the quality of each response. Then the generation policy $\pi$ is optimized by maximizing:

$$\mathbb{E}_{s \sim \mathcal{D}} \left[ \mathbb{E}_{\pi} \left[ r(a, s) \right] - \beta \mathbb{D} \left( \pi(\cdot|s) || \pi_{\text{ref}}(\cdot|s) \right) \right], \quad (1)$$

where $\mathcal{D}$ is the dataset, $\mathbb{D}$ is a divergence metric constraining the deviation from the reference policy $\pi_{\text{ref}}$, and $\beta$ is the regularization coefficient.

**Options for Constraints.** The dominant choice in existing literature is the reverse KL divergence ($\mathbb{D} = \mathbb{D}_{KL}$):

$$\mathbb{D}_{KL}(\pi(\cdot|s) || \pi_{\text{ref}}(\cdot|s)) = \sum_a \pi(a|s) \log \frac{\pi(a|s)}{\pi_{\text{ref}}(a|s)}, \quad (2)$$

which is adopted in methods such as PPO (Achiam et al., 2023), GRPO (Shao et al., 2024), and DPO (Rafailov et al.,

2023). The KL divergence attains its minimum of zero when the two distributions are identical, but diverges to positive infinity when the reference probability of an action approaches zero while the policy probability does not. This causes excessive constraints in certain scenarios.

Another constraint we would like to introduce is the L1-norm constraint, expressed as

$$\|\pi(\cdot|s) - \pi_{\text{ref}}(\cdot|s)\|_1 = \sum_a |\pi(a|s) - \pi_{\text{ref}}(a|s)|, \quad (3)$$

the advantages of which will be elucidated in Section 4.

## 4. Breaking the KL Limitations with L1-norm

### 4.1. The KL Limitations in RL for LLMs

We can identify the limitations from the solution policy under KL constraint (Ethayarajh et al., 2024; Liu et al., 2024; Rafailov et al., 2023; Azar et al., 2024; Han et al., 2025):

$$\pi_{KL}^*(a|s) = \frac{\exp\left(\frac{r(a,s)}{\beta}\right)}{\mathbb{E}_{\pi_{\text{ref}}}\left[\exp\left(\frac{r(a',s)}{\beta}\right)\right]} \pi_{\text{ref}}(a|s). \quad (4)$$

Considering the characteristics of LLMs, two critical limitations arise from Equation 4: **(1) Refuse new actions.** The probability of an action increases proportionally to its value in the reference policy ($\pi_{KL}^*(a|s) \propto \pi_{\text{ref}}(a|s)$). This multiplicative scaling imposes a significant limitation to promoting high-reward actions that have low prior probabilities ($\pi_{\text{ref}}(a|s) \to 0$). Such scenarios are ubiquitous in LLMs, for example, when the likelihood of correctly answering challenging questions approaches zero. **(2) Lack of robustness.** As indicated by Equation 4, the probability of every action shifts ($\pi_{KL}^*(a|s) \neq \pi_{\text{ref}}(a|s)$) as long as its exponentiated reward deviates from the expected value ($\exp\left(\frac{r(a,s)}{\beta}\right) \neq \mathbb{E}_{\pi_{\text{ref}}}\left[\exp\left(\frac{r(a',s)}{\beta}\right)\right]$). Furthermore, the magnitude of this shift is driven by the absolute reward value. Consequently, reward noise in any specific action, such as an inaccurate reward-model output for a particular response, can induce probability fluctuations across the entire vocabulary by perturbing the partition function $\mathbb{E}_{\pi_{\text{ref}}}\left[\exp\left(\frac{r(a',s)}{\beta}\right)\right]$, thereby significantly destabilizing post-training.

The limitations of KL divergence indicate that it is not the optimal choice for RL in LLMs, and a better constraint function needs to be identified.

### 4.2. Theoretical Insight: L1-norm Fits LLMs Better

We first examine how the L1-norm explicitly overcomes the KL limitations through its policy update formulation, and then investigate the underlying mechanism of the constraint that drives this update.

**Substantial yet Sparse Policy Updates.** We explicitly analyze the optimal policy update formulation $\Delta\pi_{L_1}^*(a|s) = \pi_{L_1}^*(a|s) - \pi_{\text{ref}}(a|s)$, to highlight its advantages:

$$\Delta\pi_{L_1}^*(a|s) = \begin{cases} \Delta_{a,s}^+, & r(a,s) = r^+ \\ -\Delta_{a,s}^-, & r(a,s) = r^+ - 2\beta \\ -\pi_{\text{ref}}(a|s), & r(a,s) < r^+ - 2\beta \\ 0, & otherwise \end{cases}, \quad (5)$$

where $r^+ = \max_{a'} r(a', s)$ denotes the maximum reward, $r^+ - 2\beta$ serves as the reward threshold determining whether action probabilities are updated (derived in Appendix A.1.1), and the terms $\Delta_{a,s}^+, \Delta_{a,s}^-$ represent probability shifts that must satisfy the mass conservation constraint:

$$\sum_{r(a,s)=r^+} \Delta_{a,s}^+ = \sum_{r(a,s)=(r^+-2\beta)} \Delta_{a,s}^- \\ + \sum_{r(a,s)<(r^+-2\beta)} \pi_{\text{ref}}(a|s). \quad (6)$$

Equations 5 and 6 imply a clear update method: $\pi_{L_1}^*$ reallocates all probability mass from actions falling below the reward threshold ($r(a, s) < r^+ - 2\beta$), and a portion ($\Delta_{a,s}^-$) from those exactly at the threshold, to the optimal actions ($r(a, s) = r^+$), while keeping the probabilities of intermediate actions unchanged. The constraint coefficient $\beta$ effectively controls this threshold, decreasing $\beta$ causes more low-reward actions to be scavenged. Notably, for actions exactly at the threshold boundary ($r(a, s) = r^+ - 2\beta$), the optimal update is non-unique, any valid reduction $\Delta_{a,s}^-$ in probability is permissible because the marginal gain in expected reward is exactly offset by the L1 penalty.

This update circumvents the KL limitations: **(1) Substantial on low-prior actions.** Unlike $\pi_{KL}^*$'s multiplicative scaling, the L1 update is additive. The probability increase depends on the regularization budget $\beta$, not on the original $\pi_{\text{ref}}$. This allows the model to instantly promote a high-reward token even if its initial probability was negligible, enabling greater policy improvement. **(2) Sparse ensuring robustness to reward noise.** The policy update is sparse and driven by the relative rank of rewards rather than their absolute magnitudes. Therefore, small reward noise that perturbs the reward values but does not alter the ranking will not cause unnecessary policy drift.

**Sparsity-inducing Constraint Mechanism.** The L1 constraint promotes sparse yet substantial policy updates by imposing a distinct penalty profile compared to KL divergence. Intuitively, L1-norm applies a stricter penalty on small probability shifts (enforcing sparsity) while being more permissive towards large deviations (inducing large updates). We formalize this sparsity-inducing mechanism in the following theorem.

**Theorem 4.1.** *Consider the optimization problem in Equation 1 with $\beta > 0$, the update under the KL constraint*

*with coefficient $\beta_{KL}$ can be reformulated as the uncon-strained maximization of the effective reward $\hat{r}_{KL} = r - \beta_{KL} \log \frac{\pi(a|s)}{\pi_{ref}(a|s)}$. Similarly, the L1-constrained up-date with coefficient $\beta_{L_1}$ corresponds to the effective reward $\hat{r}_{L_1} = r - \beta_{L_1} \text{sgn}(\pi(a|s) - \pi_{ref}(a|s))$. Let the policy perturbation magnitude be defined as $\xi = \max\left\{\frac{\pi(a|s)}{\pi_{ref}(a|s)}, \frac{\pi_{ref}(a|s)}{\pi(a|s)}\right\} - 1$. For any perturbation thresh-old $\kappa$, there always exists a coefficient $\beta_{L_1} = \beta_{KL} \log(1 + \kappa)$ such that: (1) If $\xi \leq \kappa$, then $|\hat{r}_{KL} - r| \leq |\hat{r}_{L_1} - r|$ (L1 imposes a stronger penalty); (2) If $\xi > \kappa$, then $|\hat{r}_{KL} - r| > |\hat{r}_{L_1} - r|$ (L1 imposes a weaker penalty).*

Theorem 4.1 mathematically confirms that L1-norm im-poses a stricter constraint on small perturbations ($\xi \leq \kappa$) while exerting a weaker constraint on large shifts ($\xi > \kappa$). Consequently, for actions with intermediate rewards (whose advantages are insufficient to justify large perturbations), updates are effectively sparsified by the stricter penalty; conversely, actions with extreme rewards (which warrant large shifts) encounter reduced resistance and are induced to undergo substantially larger probability updates.

## 5. Magnitude-Regularized Policy Optimization

### 5.1. The MRPO Framework

**Unbiased Estimation of the L1-norm.** Calculating the exact L1 distance between the policy $\pi$ and the reference $\pi_{\text{ref}}$ requires summing over the entire action space, which is computationally intractable for LLMs. Analogous to the estimation of KL divergence in GRPO (Shao et al., 2024), we employ an importance sampling approach to construct an unbiased estimator:

$$\|\pi(\cdot|s) - \pi_{\text{ref}}(\cdot|s)\|_1 = \mathbb{E}_{\pi_{\text{old}}}\left[\left|\frac{\pi(a|s)}{\pi_{\text{old}}(a|s)} - \frac{\pi_{\text{ref}}(a|s)}{\pi_{\text{old}}(a|s)}\right|\right],$$ 

(7)

where $\pi_{\text{old}}$ is the rollout policy. This formulation allows us to estimate the L1 penalty and its gradient unbiasedly using Monte Carlo samples from the current batch.

**Policy Optimization under L1 Constraint.** We first propose the Magnitude-Regularized Policy Optimization (MRPO) algorithm suitable for online scenarios. By simply replacing the KL divergence in token-level GRPO (Yu et al., 2026), we can derive the objective function for MRPO-G:

$$\mathcal{L}_{MRPO-G} = \frac{1}{N}\sum_i \min\left(\omega_i A_i, \text{clip}(\omega_i, \tau_l, \tau_h)A_i\right) - \beta\|\pi - \pi_{\text{ref}}\|_1,$$

(8)

where, $\omega_i = \frac{\pi(a_i|s_i)}{\pi_{\text{old}}(a_i|s_i)}$, and $A_i$ is the advantage function estimated using the same method as in GRPO. $a_i$ and $s_i$ denote the $i$-th token and its context, $\pi$ is the current policy of the LLM, $\pi_{\text{old}}$ is the rollout policy, and $\pi_{\text{ref}}$ is the refer-

ence policy. $N$ is the total number of tokens in the batch. $\tau_l$ and $\tau_h$ are two hyperparameters used to prevent variance spikes caused by $\omega_i$, and $\beta$ is a hyperparameter controlling the constraint strength.

In addition, we also experimented with applying the L1-norm to the REINFORCE (Williams, 1992) algorithm and incorporated a clipping mechanism to enhance stability:

$$\mathcal{L}_{MRPO-R} = \frac{1}{N}\sum_i \min\left(g_i A_i, \text{clip}(g_i, \tau_l', \tau_h')A_i\right) - \beta\|\pi - \pi_{\text{ref}}\|_1.$$

(9)

where $g_i = \log(\omega_i)$, $\tau_l' = \log(\tau_l)$, $\tau_h' = \log(\tau_h)$, and all other parts remain identical to those defined in MRPO-G.

**Extending to Offline Scenarios** Given that the theoreti-cal assumptions of MRPO do not require online learning, MRPO should also be applicable in offline settings. How-ever, in offline settings, since the rollout policy $\pi_{\text{old}}$ and the current policy $\pi$ may differ significantly, this can lead to nu-merical overflow issues in $\omega_i$ and $g_i$. Therefore, we need to reduce the gradient in these cases to avoid instability. Specif-ically, we introduce a dynamically adjusted coefficient $\lambda_i$, to the expected reward term to ensure gradient stability:

$$\mathcal{L}'_{MRPO-G} = \frac{1}{N}\sum_i \lambda_i \min\left(\omega_i A_i, \text{clip}(\omega_i, \tau_l, \tau_h)A_i\right) - \beta\|\pi - \pi_{\text{ref}}\|_1,$$

(10)

$$\mathcal{L}'_{MRPO-R} = \frac{1}{N}\sum_i \lambda_i \min\left(g_i A_i, \text{clip}(g_i, \tau_l', \tau_h')A_i\right) - \beta\|\pi - \pi_{\text{ref}}\|_1,$$

(11)

where $\lambda_i$ is defined as a function of the estimated L1-norm:

$$\lambda_i = \frac{1}{1 + \frac{\alpha}{|G_i|}\|\pi - \pi_{\text{ref}}\|_1}$$

(12)

where $G_i$ denotes the group containing the $i$-th token, i.e., the set of all tokens that are grouped together with token $i$. The group definition follows that in GRPO. Alternatively, $\lambda_i$ can be interpreted in another way. When the constrained optimization form is used as the objective function and the learning rate with $\alpha\beta$ as the Lagrange multiplier is applied, Equations 10 and 11 are equivalent to the dual gradient descent, with the model's parameter learning rate being $\lambda_i$ (proof provided in Appendix A.5).

### 5.2. Theoretical Analysis

**Trust-Region Analysis.** Since KL divergence is widely used to keep policy deviation within a trust region and thereby ensure convergence (Schulman et al., 2015; 2017), this naturally raises the question of whether the L1-norm can provide similar control. While TRPO (Schulman et al.,

2015) establishes a guarantee using the worst-case TV divergence (equivalently the maximum L1-norm up to a constant factor), the original analysis does not directly provide a guarantee in terms of the expected L1-norm used in MRPO, which is generally less conservative and more practical. We answer this affirmatively by establishing a TRPO-style monotonic improvement bound with an expected L1-norm penalty, and provide in Appendix A.2 a proof based on the recursive property of MDPs, which differs from the proof route used in TRPO.

**Theorem 5.1.** *Let $\pi_0$ be the current policy and $\pi$ be the updated policy. Under standard regularity assumptions and bounded rewards, the expected discounted reward $\eta(\pi)$ is lower-bounded by the surrogate objective penalized by the L1-norm: $\eta(\pi) \geq L_{\pi_0}(\pi) - C \cdot \sum_t \gamma^t \frac{1+\gamma-\gamma^2}{1-\gamma} \mathbb{E}_{p(s_t|\pi_0)} [\|\pi(\cdot|s_t) - \pi_0(\cdot|s_t)\|_1]$, where $L_{\pi_0}(\pi)$ is the standard local approximation to $\eta(\pi)$ and $C$ is a constant related to the reward magnitude.*

Theorem 5.1 shows that the true expected return $\eta(\pi)$ can be lower-bounded by an L1-penalized surrogate objective, so improving this lower bound yields a TRPO-style monotonic-improvement guarantee. This confirms that the expected L1-norm can serve as a sufficient trust-region control term for monotonic improvement, thereby supporting training stability. However, since MRPO also retains clipping techniques to control policy deviation in practice, this theorem should be interpreted as a trust-region justification for the L1 regularizer, rather than as a direct convergence proof of the full clipped MRPO objective.

**Contraction Toward the Optimal Policy.** Beyond stability, we analyze the efficiency of policy improvement. We quantify the improvement using the contraction coefficient $k(\pi) = \frac{d(\pi, \pi^*)}{d(\pi_{\text{ref}}, \pi^*)}$, which measures the progress of the updated policy $\pi$ towards the global optimum $\pi^*$, relative to the initial discrepancy from $\pi_{\text{ref}}$, under the discrepancy measure $d \in \{L_1, \mathbb{D}_{KL}\}$. A smaller $k$ indicates a larger relative progress towards $\pi^*$, and our theoretical analysis shows that the L1 constraint allows for more substantial policy improvement in certain extreme cases.

**Theorem 5.2.** *Assume rewards are bounded, $\pi_{ref}$ does not approach $\pi^*$, and $\beta > 0$. Let $\pi^*$ denote the policy that maximizes the expected reward and $a^*$ be the action with the highest reward. In scenarios where $\pi_{ref}(a^*|s) \to 0$ or $\exp(\frac{r(a^*|s)}{\beta}) \to \mathbb{E}_{\pi_{ref}} \left[ \exp(\frac{r(a,s)}{\beta}) \right]$, it holds that $k(\pi_{KL}^*) \to 1$, whereas $k(\pi_{L_1}^*)$ remains bounded away from 1. Crucially, in the specific case where $\pi_{ref}(a^*|s) \to 0$ and $d = \mathbb{D}_{KL}$, we achieve $k(\pi_{L_1}^*) \to 0$.*

Theorem 5.2 demonstrates that the policy optimized with the L1-norm can make more substantial progress in certain extreme cases. For example, when $\pi_{\text{ref}}(a^*|s) \to 0$, the policy

updated under the KL constraint only moves slightly toward the optimal policy ($k \to 1$), while the L1-norm does not suffer from this issue. In such cases, regardless of how the constraint coefficient $\beta > 0$ is chosen, the L1-regularized objective allows a more substantial update, thereby providing an advantage in this regime. Although the theorem only guarantees the advantage in extreme cases, such cases are common in LLM settings: an LLM may fail to produce the correct answer even after thousands of attempts, meaning that the probability of the optimal response under the reference model is close to zero. This suggests that, compared to KL regularization, the L1 constraint can be particularly suitable for LLMs.

# 6. Experiments

## 6.1. Experimental Setup

We begin by validating MRPO across diverse scenarios, including preference alignment, reasoning, and offline RL, to comprehensively demonstrate its effectiveness. To test robustness, we further evaluate MRPO using a weaker reward model. To investigate the theoretical mechanism of the L1 constraint, we compare policy behaviors under different regularizers and examine the correlation between the L1 norm and KL divergence. Finally, we analyze response distributions to demonstrate MRPO's ability to learn low-probability optimal actions.

**Preference Alignment.** Following standard practices (Achiam et al., 2023; Li et al., 2023; Ahmadian et al., 2024), we utilize the Ultrafeedback dataset (Cui et al., 2023) for online reinforcement learning, primarily employing a 3B reward model (Yang et al., 2024) that balances alignment accuracy with computational efficiency. To further assess robustness against noisy signals, we also train Qwen2.5-7B-Instruct (Bai et al., 2023) using a weaker 2B reward model. Our extensive evaluation spans four diverse architectures to validate MRPO's effectiveness: Meta-Llama3.1-8B-Instruct (Dubey et al., 2024), Mistral-7B-Instruct-v0.3 (Jiang et al., 2023), and Qwen2.5-Instruct (1.5B and 3B) (Bai et al., 2023). We assess preference alignment on AlpacaEval (Dubois et al., 2024) and Arena-Hard (Li et al., 2024), utilizing the official Meta-Llama-3-70B-Instruct judge for AlpacaEval (reporting $WR_1$, $WR_2$, and $LC_2$) and the GRM-Llama3.1-8B reward model for Arena-Hard (calculating win rates against GPT-4-0613 and Gemini-2.0-flash). We compare MRPO against established online baselines, including KL-based PPO, GRPO, BNPO, DPO and IPO, and DPH with other constraints. All methods are implemented using the TRL library (von Werra et al., 2020), with detailed hyperparameter configurations provided in Appendix B.1.

*Table 1.* Experimental results on preference alignment. Here, "Base" refers to the metrics of the base model (Llama or Mistral). The "Constraint" column lists the constraint terms: R-KL/F-KL for reverse/forward KL divergence, JS for Jensen-Shannon divergence, and L1 for L1-norm. **Bold** and underlined indicate the best and second-best performance. On Alpaca Eval, $WR_1$ and $WR_2$ denote the Win Rates judged by Llama3-70B-Instruct on versions 1.0 and 2.0, respectively; $LC_2$ refers to the Length-Controlled Win Rates on v2.0. For Arena Hard, we report Win Rates against GPT-4-0613 (v0.1) and Gemini-2.0-Flash (v2.0), denoted as GPT-4 and Gemini-2, respectively.

| Constraint | Method | Llama3.1-8B-Instruct | | | | | Mistral-7B-Instruct | | | | |
| | | Alpaca Eval | | | Arena Hard | | Alpaca Eval | | | Arena Hard | |
| | | $WR_1 \uparrow$ | $WR_2 \uparrow$ | $LC_2 \uparrow$ | GPT-4 $\uparrow$ | Gemini-2 $\uparrow$ | $WR_1 \uparrow$ | $WR_2 \uparrow$ | $LC_2 \uparrow$ | GPT-4 $\uparrow$ | Gemini-2 $\uparrow$ |
|---|---|---|---|---|---|---|---|---|---|---|---|
| – | Base | 92.29 | 42.13 | 46.72 | 36.20 | 17.92 | 93.11 | 18.52 | 27.61 | 25.40 | 13.50 |
| R-KL | PPO | 94.52 | 45.32 | 52.50 | 49.80 | 20.19 | 91.64 | 18.59 | 26.23 | 26.20 | 11.36 |
| | DPO | 95.30 | 48.54 | 55.78 | 42.20 | 19.65 | 92.75 | 23.25 | 30.59 | 30.80 | 15.64 |
| | IPO | 94.89 | 48.57 | 54.53 | 41.20 | 19.79 | 92.33 | 22.10 | 29.92 | 32.40 | 16.04 |
| | f-PO | 93.05 | 42.82 | 48.83 | 38.20 | 17.78 | 91.67 | 18.45 | 25.78 | 26.00 | 13.90 |
| | GRPO | 97.34 | 53.97 | 60.51 | 61.80 | 32.22 | 93.33 | 29.50 | 38.13 | 39.20 | 17.11 |
| | BNPO | 96.54 | 52.47 | 60.37 | 59.00 | 25.94 | 93.76 | 31.17 | 36.26 | 38.00 | 17.65 |
| | Dr-GRPO | 97.17 | 51.32 | 62.49 | 62.00 | 29.01 | 93.54 | 29.59 | 36.16 | 33.60 | 16.31 |
| F-KL | DPH-F | 96.77 | 48.89 | 56.43 | 60.80 | 27.14 | 93.23 | 27.64 | 35.59 | 39.00 | 17.11 |
| JS | DPH-JS | 97.30 | 50.17 | 57.90 | 62.80 | 31.68 | 93.09 | 27.54 | 34.29 | 35.20 | 15.78 |
| L1 | MRPO-G | 97.59 | 67.73 | 73.09 | 94.40 | 81.82 | 94.36 | 43.45 | **47.48** | 59.60 | **31.02** |
| | MRPO-R | **98.34** | **69.20** | **76.15** | **96.00** | **84.09** | **94.73** | **46.21** | 47.14 | **63.00** | 29.68 |

*Table 2.* RLVR experimental results using Qwen2.5-7B as the base model. The metric pass@$k$ denotes the proportion of problems for which at least one correct solution is generated out of $k$ attempts under identical sampling conditions. **Bold** and underlined indicate the best and second-best performance.

| Method | GSM8k | Code-R1 | | |
| | pass@1 | pass@1 | pass@3 | pass@5 |
|---|---|---|---|---|
| Base | 62.44 | 4.92 | 5.76 | 6.18 |
| REINFORCE++ | 79.38 | 11.52 | 13.90 | 14.47 |
| GRPO | 77.41 | 26.97 | 31.18 | 32.72 |
| DAPO | 83.09 | 13.62 | 16.29 | 17.42 |
| BNPO | 79.98 | 8.15 | 9.55 | 9.97 |
| Dr-GRPO | 77.63 | 31.60 | 38.20 | 40.45 |
| DPH-F | 80.74 | 17.56 | 20.93 | 21.63 |
| DPH-JS | 75.36 | 22.19 | 25.98 | 27.39 |
| MRPO-G | **84.08** | **33.56** | **39.75** | 40.73 |
| MRPO-R | 83.32 | 33.29 | 39.47 | **41.29** |

**Reasoning.** We applied KW-R1 (Liang et al., 2025) to build the RLVR framework, using the correctness of the model's reasoning outputs as the reward signal for training Qwen2.5-7B (Bai et al., 2023) base model. For the math task, we trained the model on the GSM8k (Cobbe et al., 2021) training set, where it was prompted to generate its own chain-of-thought marked within thinking tags, and rewards were assigned based on whether its final answer was correct. The model's performance was then assessed on the GSM8k test split using the pass@1 metric. For the coding task, training was conducted on the Code-R1 (Liu & Zhang, 2025) dataset. Rewards were computed according to multiple factors: whether the code produced after reasoning executed successfully, passed all tests, returned correct out-

puts, and met predefined limits on memory consumption and execution time. Evaluation on the Code-R1 test set used pass@1, pass@3, and pass@5 to capture the fraction of problems solved across several reasoning attempts. Additional information on training and evaluation procedures can be found in Appendix B.2.

**Offline RL.** In line with earlier studies (Ethayarajh et al., 2024), we used the UltraFeedback (Cui et al., 2023) dataset to perform offline training of the LLM and evaluated the model on MMLU (Hendrycks et al., 2020), GSM8k (Cobbe et al., 2021), PIQA (Bisk et al., 2020), GPQA (Rein et al., 2023), ARC-Easy (ARC-E), and ARC-Challenge (ARC-C) (Clark et al., 2018). Specifically, PIQA, ARC-E, and MMLU primarily assess fundamental general capabilities, whereas ARC-C, GSM8k, and GPQA evaluate the model's complex reasoning abilities. To facilitate a structured comparison, we report the average score of benchmarks within each category, and the mean score across all benchmarks as the comprehensive overall indicator. Because these benchmarks rely heavily on factual knowledge and the correctness ensured by human annotation, something reward models cannot consistently match, we directly adopted the original data pairs without generating new samples or re-assigning labels. To maintain fairness with baselines that do not use reward magnitudes, MRPO assigns +1 to positive examples and -1 to negative ones as rewards. Training was implemented using TRL, and evaluations on MMLU and the remaining benchmarks were performed through llama-factory (Zheng et al., 2024) and evalscope (Team, 2024). Additional training details are provided in Appendix B.3.

*Table 3.* Experimental results on Offline RL. We report aggregated scores across general capabilities (average score of PIQA, ARC-E and MMLU) and complex reasoning capabilities (average score of ARC-C, GSM8k and GPQA). Average denotes the mean score across all individual benchmarks. **Bold** indicates the best performance, and underlined denotes the second best.

| Method | Gemma-7b-it | | | Llama2-7b-chat-hf | | |
|---|---|---|---|---|---|---|
| | General ↑ | Reasoning ↑ | Average ↑ | General ↑ | Reasoning ↑ | Average ↑ |
| Base | 60.53 | 44.09 | 52.31 | 49.91 | 26.08 | 38.00 |
| DPO | 60.82 | 44.61 | 52.72 | 49.80 | 26.54 | 38.17 |
| IPO | 60.96 | 44.20 | 52.58 | 49.97 | 27.13 | 38.55 |
| SLiC | 60.86 | 43.81 | 52.33 | 49.89 | 26.70 | 38.30 |
| RPO | 60.87 | 44.12 | 52.50 | 49.64 | 27.19 | 38.42 |
| BCO | 60.87 | 43.68 | 52.28 | 49.91 | 26.70 | 38.31 |
| simPO | 60.74 | 44.21 | 52.47 | 49.80 | 27.16 | 38.48 |
| MRPO-G | 60.99 | **45.40** | **53.20** | **50.07** | **27.44** | **38.75** |
| MRPO-R | **61.04** | 44.74 | 52.89 | 50.00 | 27.34 | 38.67 |

## 6.2. Experimental Results

**Preference Alignment Results.** The results of the preference alignment experiment are shown in Table 1, where we list the preference alignment scores of models trained with different algorithms. It can be seen that MRPO outperforms all baseline methods across all benchmarks. Specifically, MRPO-R outperforms the best baseline by 129% and 87% in the gains of $WR_2$ and $LC_2$ on the Alpaca-Eval benchmark with Llama3.1. On the win rates against GPT-4 and Gemini-2 on Arena-Hard, these multipliers increase to a remarkable 125% and 363%. The performance of MRPO on Mistral is equally impressive. On Alpaca-Eval, MRPO achieves improvements of 2.19 times and 1.89 times the best baseline in $WR_2$ and $LC_2$ scores, respectively, while on Arena-Hard, it shows multipliers of 2.72 and 4.22. Notably, MRPO achieves more significant gains on metrics where the base model scores are lower (e.g., $WR_2$, Gemini-2). We attribute this to the fact that challenging tasks necessitate the acquisition of new actions, a scenario that perfectly exploits the primary advantage of the L1 constraint over KL.

**RLVR Results.** Table 2 reports the results on mathematical reasoning (GSM8k) and code generation (Code-R1). MRPO consistently outperforms all baseline methods across both benchmarks, demonstrating its robustness in reasoning-intensive scenarios. On the relatively easier GSM8k dataset, MRPO-G achieves a pass@1 of 84.08%, surpassing the strong baseline DAPO (83.09%). More importantly, the superiority of MRPO is substantially amplified on the more challenging Code-R1 task, where the base model exhibits extremely low performance (4.92%). In this high-difficulty regime, MRPO-G achieves a pass@1 of 33.56%, significantly outperforming GRPO (26.97%) and even the strong variant Dr-GRPO (31.60%). This empirical evidence strongly aligns with our theoretical analysis: while KL-based methods (e.g., GRPO) struggle to promote low-probability optimal actions due to excessive penalties, the

*Table 4.* Preference alignment results using Qwen2.5-7B-Instruct with a weaker reward model. The "Constraint" column lists the regularization terms: R-KL/F-KL for reverse/forward KL divergence, JS for Jensen-Shannon divergence, and L1 for L1-norm. Here, $WR_1$ and $WR_2$ denote the Win Rates on versions 1.0 and 2.0 of AlpacaEval, respectively, while $LC_2$ refers to the Length-Controlled Win Rate on version 2.0. **Bold** and underlined indicate the best and second-best performance.

| Constraint | Method | Alpaca Eval | | |
|---|---|---|---|---|
| | | $WR_1$ ↑ | $WR_2$ ↑ | $LC_2$ ↑ |
| – | Base | 93.23 | 37.33 | 44.67 |
| R-KL | PPO | 94.31 | 36.28 | 45.22 |
| | DPO | 93.45 | 38.13 | 45.42 |
| | IPO | 93.77 | 39.37 | 47.50 |
| | GRPO | 94.79 | 39.62 | 45.24 |
| | BNPO | 94.42 | 40.13 | 46.54 |
| | Dr-GRPO | 93.68 | 39.89 | 45.37 |
| F-KL | DPH-F | 94.41 | 40.24 | 46.94 |
| JS | DPH-JS | 93.63 | 39.96 | 45.97 |
| L1 | MRPO-G | **96.09** | **48.51** | 51.45 |
| | MRPO-R | 95.69 | 48.24 | **53.00** |

sparsity-inducing nature of the L1 constraint allows MRPO to aggressively boost the probability of correct reasoning paths in hard tasks, leading to substantial performance leaps.

**Offline RL Results.** Table 3 summarizes the evaluation results in the offline RL scenario. MRPO consistently achieves state-of-the-art performance, surpassing all strong baselines in terms of average scores across both models. Most importantly, MRPO exhibits a particularly pronounced advantage in complex reasoning capabilities compared to general tasks. While KL-based methods often struggle to significantly improve reasoning performance over the base model (e.g., on Gemma, DPO yields a modest gain of +0.52 in the reasoning category), MRPO-G delivers a substantial boost (improving by +1.31 to reach 45.40). In contrast, the performance gaps

*Table 5.* Preference alignment results on Alpaca Eval across models of varying parameter sizes. $WR_1$ and $WR_2$ denote the Win Rates on versions 1.0 and 2.0, respectively, while $LC_2$ refers to the Length-Controlled Win Rate on version 2.0. **Bold** indicates the best performance, and underlined denotes the second best.

| Method | Qwen2.5-1.5B-Instruct Alpaca Eval | | | Qwen2.5-3B-Instruct Alpaca Eval | | |
|---|---|---|---|---|---|---|
| | $WR_1$ | $WR_2$ | $LC_2$ | $WR_1$ | $WR_2$ | $LC_2$ |
| Base | 55.94 | 8.16 | 8.41 | 87.24 | 35.78 | 34.12 |
| GRPO | 66.69 | 17.00 | 12.62 | 88.74 | 49.10 | 44.84 |
| BNPO | 72.42 | 22.53 | 16.61 | 91.04 | 51.70 | 46.18 |
| Dr-GRPO | 72.85 | 22.76 | 17.26 | 90.60 | 50.41 | 44.41 |
| MRPO-G | **77.07** | **28.46** | **21.24** | **93.13** | **62.25** | **56.21** |
| MRPO-R | 75.52 | 27.51 | 20.67 | 91.49 | 57.64 | 54.97 |

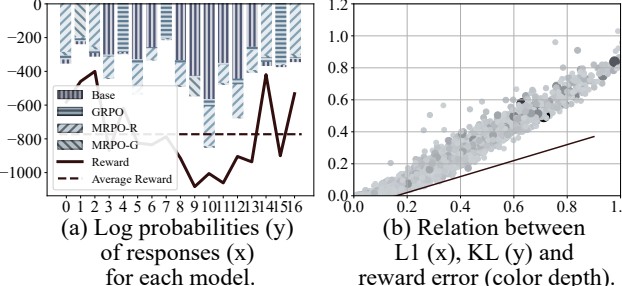

(a) Log probabilities (y) of responses (x) for each model.

(b) Relation between L1 (x), KL (y) and reward error (color depth).

*Figure 2.* Policies obtained by different algorithms and the relationships between KL divergence, L1 norm, and reward error, where darker colors indicate greater errors.

on general tasks are relatively narrow across all methods. This disparity suggests that while standard KL constraints are sufficient for maintaining general language capabilities, the sparsity-inducing L1 constraint of MRPO is critical for unlocking superior reasoning abilities, where precise policy adjustments are required to navigate complex logical paths.

**Results under Weak Reward Model.** Table 4 presents the experimental results in the weak reward model scenario. Attributable to the noise inherent in the rewards, most algorithms fail to yield meaningful improvements; notably, PPO even regresses below the base model on the $WR_1$ metric. In stark contrast, MRPO exhibits exceptional robustness, achieving dramatic performance surges that outperform the best baseline by factors of $3.84\times$, $2.94\times$, and $1.81\times$ on $WR_2$, $LC_2$, and $WR_1$, respectively. This empirical evidence confirms that MRPO can effectively mitigate the impact of reward noise, securing significant policy improvements even when the reward supervision is highly unreliable.

**Results Across Model Sizes.** Table 5 evaluates the scalability of MRPO across models with varying parameter counts. MRPO consistently dominates all baseline methods across both model scales, demonstrating its robustness re-

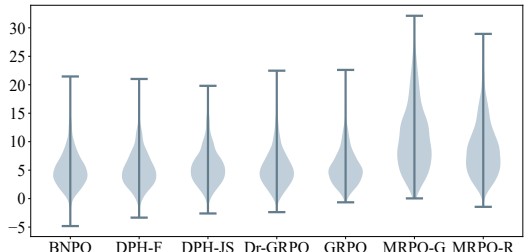

*Figure 3.* Distribution of rewards for responses from models trained with different algorithms.

gardless of model capacity. On the 1.5B model, where the base performance is relatively limited ($WR_2$ 8.16%), MRPO-G achieves a remarkable improvement, boosting $WR_2$ to 28.46% and significantly outperforming the strongest baseline, Dr-GRPO (22.76%). This trend of superiority is maintained and even amplified on the 3B model, where MRPO-G attains a $WR_2$ of 62.25%, establishing a substantial lead of over 10 points against BNPO (51.70%). Furthermore, the consistent gains in the Length-Controlled metric ($LC_2$) confirm that MRPO's improvements stem from genuine quality enhancements rather than mere length exploitation, validating its effectiveness as a scalable alignment solution.

**Constraint Behavior.** To investigate the mechanism of L1 regularization, we visualize the response probabilities in Figure 2(a) and the relationship between different policy divergence metrics in Figure 2(b), where color intensity represents the reward error (measured by the inconsistency between 3B and 8B reward models). Figure 2(a) corroborates the prediction in Figure 1: the MRPO policy exhibits sparse yet substantial updates. Specifically, compared to GRPO, MRPO applies significantly larger probability boosts to high-advantage responses (e.g., indices 0, 1, 2) and deeper suppression to low-reward ones. Crucially, for actions with mediocre rewards (e.g., 4, 7, 15), the policy remains strictly aligned with the reference, unlike GRPO which introduces unnecessary shifts. Turning to Figure 2(b), all data points lie above a linear bound, indicating that the L1 penalty grows significantly slower than the KL constraint. Furthermore, while the reward error increases with both the L1-norm and KL divergence, it does not exhibit a distinct preference for either metric. In contrast, the region where KL diverges significantly but L1 remains moderate contains almost no high-error points. This suggests that the L1-norm is sufficient for controlling reward hacking, rendering the excessively restrictive KL penalty unnecessary.

**Distribution of Response.** Figure 3 illustrates the reward distributions of responses generated by LLMs trained with different algorithms. It is evident that LLMs trained via MRPO exhibit a significantly broader reward distribution

compared to baselines. Crucially, in the high-reward regime (e.g., $> 15$), where the generation probabilities of other algorithms vanish to nearly zero, MRPO maintains a viable probability mass. This observation empirically confirms that MRPO successfully learns to promote high-reward actions with low priors, overcoming the exploration limitations that restrict other methods. Moreover, the broader reward distribution also indicates greater rollout diversity, suggesting that MRPO does not suffer from entropy collapse.

## 7. Conclusions, Limitations, and Future Work

In this paper, we analyze the different effects of several regularization constraints on reinforcement learning for large language models, and introduce **M**agnitude-**R**egularized **P**olicy **O**ptimization (MRPO), an RL algorithm based on the L1 constraint. Theoretically, we prove that MRPO enables substantial probability boosts for low-prior optimal actions while maintaining robustness against reward noise. Empirically, MRPO outperforms GRPO in preference alignment, surpasses DAPO in reasoning tasks, and exceeds DPO in offline RL using only binary rewards, demonstrating its general effectiveness.

However, this paper analyzes only a limited set of regularization constraints, such as the L1-norm and KL divergence. A promising future work is to study the behavior of more constraints in a broader space of discrepancy measures and developing a more rigorous theoretical framework that better aligns with practical applications. In addition, experiments in more scenarios, such as broader language tasks, agentic reinforcement learning, and multimodal reinforcement learning, would also be of significant value.

## Acknowledgements

The authors would like to thank all the anonymous reviewers for their insightful comments. This work is supported by the National Science and Technology Major Project (No. 2025ZD1606200 and Sub-project No. 2025ZD1606203) and the National Natural Science Foundation of China (No. 92470205).

## Impact Statement

This paper aims to contribute to the advancement of reinforcement learning for Large Language Models, including but not limited to RLHF, RLVR, and offline RL. Although this work is primarily methodological, improving RL-based LLM post-training may affect the reliability and behavior of deployed language models. Potential risks include over-optimization toward imperfect rewards and unintended changes in model behavior. We encourage careful evaluation before deployment.

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

# A. Proofs

## A.1. Proofs for Section 4.1

### A.1.1. OPTIMAL POLICY UNDER L1 NORM CONSTRAINT

*Proof.* For convenience, the state $s$ is omitted in the following derivations. When the constraint $\mathbb{D}$ is the L1-norm, Equation 1 can be expressed as:

$$\mathbb{E}_\pi\left[r(a)\right] - \beta\left\|\pi - \pi_{\text{ref}}\right\|_1, \tag{13}$$

where the target policy $\pi$ must satisfy $\sum_a \pi(a) = 1$ and $0 \le \pi(a) \le 1$ for any $a$.

When the objective function is optimal, the solved policy $\pi^*_{L_1}$ should satisfy the KKT conditions:

$$
\begin{aligned}
\forall a, \quad & r(a) - \beta g(a) + \lambda - \mu_1(a) + \mu_2(a) = 0, \\
& 0 \le \mu_1(a), \mu_2(a), \quad \mu_1(a)(1 - \pi^*_{L_1}(a)) = 0, \quad \mu_2(a)\pi^*_{L_1}(a) = 0.
\end{aligned} \tag{14}
$$

where $\lambda$, $\mu_1(a)$ and $\mu_2(a)$ are Lagrange multipliers, and $g(a)$ is the sub-gradient of $\|\pi - \pi_{\text{ref}}\|_1$ when $\pi = \pi^*_{L_1}$. When $\pi_{\text{ref}}$ is the optimal solution to Equation 13, the optimal policy is evidently still $\pi_{\text{ref}}$, which satisfies Equation 5. Therefore, we only need to consider the case where $\pi_{\text{ref}}$ is not the optimal policy. Clearly, at this point, unless $\pi_{\text{ref}}$ is already the optimal policy, there must exist actions whose probability increases. We proceed by focusing on such actions.

**Case 1:** $\exists a^* \ s.t. \ \pi^*_{L_1}(a^*) = 1$. At this point, the probabilities of all other actions are 0, so the KKT conditions can be simplified as:

$$
\begin{aligned}
& r(a^*) - \beta + \lambda - \mu_1(a^*) = 0, \\
& \forall \pi^*_{L_1}(a) < \pi_{\text{ref}}(a) : r(a) + \beta + \lambda + \mu_2(a) = 0
\end{aligned} \tag{15}
$$

Combining the equations and substituting into condition $\mu_1(a^*) + \mu_2(a) \ge 0$, we obtain:

$$\forall \pi^*_{L_1}(a) < \pi_{\text{ref}}(a) : \quad r(a^*) \ge r(a) + 2\beta, \tag{16}$$

which satisfies Equation 5.

**Case 2:** Otherwise. At this point, Equation 14 can be simplified to:

$$
\begin{aligned}
& \forall \pi^*_{L_1}(a^+) > \pi_{\text{ref}}(a^+) : \quad r(a^+) - \beta + \lambda = 0, \\
& \forall 0 = \pi^*_{L_1}(a^{--}) : \quad r(a^{--}) + \beta + \lambda + \mu_2(a^{--}) = 0, \\
& \forall 0 < \pi^*_{L_1}(a^-) < \pi_{\text{ref}}(a^-) : \quad r(a^-) + \beta + \lambda = 0, \\
& \forall \pi^*_{L_1}(a^0) = \pi_{\text{ref}}(a^0) : \quad r(a^0) - \beta g(a^0) + \lambda = 0.
\end{aligned} \tag{17}
$$

Given that both $\mu_1(a)$ and $\mu_2(a)$ are non-negative, and $-1 \le g(a^0) \le 1$, the final conclusion can be drawn:

$$
\begin{aligned}
& r(a^+) = r(a^-) + 2\beta, \\
& r(a^0) \le r(a^+) \le r(a^0) + 2\beta, \\
& r(a^+) \ge r(a^-) + 2\beta, \\
& r(a^{--}) \le r(a^0)
\end{aligned} \tag{18}
$$

Finally, the form of Equation 5 can be obtained.

$\square$

### A.1.2. THEOREMS

**Theorem 4.1** Consider the optimization problem in Equation 1 with $\beta > 0$, the update under the KL constraint with coefficient $\beta_{KL}$ can be reformulated as the unconstrained maximization of the effective reward $\hat{r}_{KL} = r - \beta_{KL}\log\frac{\pi(a|s)}{\pi_{\text{ref}}(a|s)}$. Similarly, the L1-constrained update with coefficient $\beta_{L_1}$ corresponds to the effective reward $\hat{r}_{L_1} = r - \beta_{L_1}\text{sgn}(\pi(a|s) - \pi_{\text{ref}}(a|s))$. Let the policy perturbation magnitude be defined as $\xi = \max\left\{\frac{\pi(a|s)}{\pi_{\text{ref}}(a|s)}, \frac{\pi_{\text{ref}}(a|s)}{\pi(a|s)}\right\} - 1$. For any perturbation threshold $\kappa$, there always exists a coefficient $\beta_{L_1} = \beta_{KL}\log(1 + \kappa)$ such that: **(1)** If $\xi \le \kappa$, then $|\hat{r}_{KL} - r| \le |\hat{r}_{L_1} - r|$ (L1 imposes a stronger penalty); **(2)** If $\xi > \kappa$, then $|\hat{r}_{KL} - r| > |\hat{r}_{L_1} - r|$ (L1 imposes a weaker penalty).

*Proof.* When the KL constraint is used, the gradient of Equation 1 is:

$$
\begin{aligned}
G_{KL}(\theta) &= \nabla_\theta \mathbb{E}_\pi \left[ r \right] - \beta_{KL} \nabla_\theta \mathbb{D}_{KL}(\pi \| \pi_{\text{ref}}) \\
&= \mathbb{E}_\pi \left[ r \nabla_\theta \log \pi(a|s) \right] - \beta_{KL} \sum_a (1 + \log \tfrac{\pi(a|s)}{\pi_{\text{ref}}(a|s)}) \nabla_\theta \pi(a|s) \\
&= \mathbb{E}_\pi \left[ r \nabla_\theta \log \pi(a|s) \right] - \beta_{KL} \sum_a \log \tfrac{\pi(a|s)}{\pi_{\text{ref}}(a|s)} \nabla_\theta \pi(a|s) - \beta_{KL} \nabla_\theta \sum_a \pi(a|s) \\
&= \mathbb{E}_\pi \left[ r \nabla_\theta \log \pi(a|s) \right] - \beta_{KL} \sum_a \pi(a|s) \log \tfrac{\pi(a|s)}{\pi_{\text{ref}}(a|s)} \nabla_\theta \log \pi(a|s) - \beta_{KL} \nabla_\theta 1 \\
&= \mathbb{E}_\pi \left[ \left( r - \beta_{KL} \log \tfrac{\pi(a|s)}{\pi_{\text{ref}}(a|s)} \right) \nabla_\theta \log \pi(a|s) \right],
\end{aligned}
\tag{19}
$$

which is identical to the gradient when using $\hat{r}_{KL}$ as the reward with no constraint. Similarly, the gradient for the L1 constraint can be derived:

$$
\begin{aligned}
G_{L_1}(\theta) &= \nabla_\theta \mathbb{E}_\pi \left[ r \right] - \beta_{L_1} \nabla_\theta \| \pi - \pi_{\text{ref}} \| \\
&= \mathbb{E}_\pi \left[ r \nabla_\theta \log \pi(a|s) \right] - \beta_{L_1} \sum_a \text{sgn}(\pi(a|s) - \pi_{\text{ref}}(a|s)) \nabla_\theta \pi(a|s) \\
&= \mathbb{E}_\pi \left[ r \nabla_\theta \log \pi(a|s) \right] - \beta_{L_1} \sum_a \pi(a|s) \, \text{sgn}(\pi(a|s) - \pi_{\text{ref}}(a|s)) \nabla_\theta \log \pi(a|s) \\
&= \mathbb{E}_\pi \left[ \left( r - \beta_{L_1} \text{sgn}(\pi(a|s) - \pi_{\text{ref}}(a|s)) \right) \nabla_\theta \log \pi(a|s) \right],
\end{aligned}
\tag{20}
$$

which is identical to the gradient when using $\hat{r}_{L_1}$ as the reward with no constraint. Substituting the conditions proves the subsequent conclusion. $\square$

## A.2. Proofs for Section 5

Following the standard reinforcement learning setup, we denote the state at time $t$ as $s_t$, the action at time $t$ as $a_t$ and the discount factor as $\gamma$. We denote $\pi_{old}$ as $\pi_0$. Similar to TRPO (Schulman et al., 2015), we first prove several lemmas, and then use these lemmas to prove the final conclusion. It is worth noting that although the structure of our proof is relatively similar to that of TRPO, we employ a proof method based on the recursive nature of Markov Decision Processes, which differs from the proof approach of TRPO. This allows us to derive a tighter bound than TRPO.

**Lemma A.1.** *Let $e_t = \sum_{s_t} |p(s_t|\pi) - p(s_t|\pi_0)|$. Then the inequality $e_t \leq \sum_{i < t} \mathbb{E}_{p(s_i|\pi_0)} \left[ \| \pi(\cdot|s_i) - \pi_0(\cdot|s_i) \|_1 \right]$ holds.*

*Proof.* Using the triangle inequality and the fact that the sum of probabilities is 1, we can obtain:

$$
\begin{aligned}
e_t &= \sum_{s_t} |p(s_t|\pi) - p(s_t|\pi_0)| \\
&\leq \sum_{s_t} \sum_{s_{t-1}} |p(s_t|s_{t-1},\pi)p(s_{t-1}|\pi) - p(s_t|s_{t-1},\pi)p(s_{t-1}|\pi_0) + p(s_t|s_{t-1},\pi)p(s_{t-1}|\pi_0) - p(s_t|s_{t-1},\pi_0)p(s_{t-1}|\pi_0)| \\
&\leq \sum_{s_t} \sum_{s_{t-1}} p(s_t|s_{t-1},\pi) |p(s_{t-1}|\pi) - p(s_{t-1}|\pi_0)| + \sum_{s_t} \sum_{s_{t-1}} p(s_{t-1}|\pi_0) |p(s_t|s_{t-1},\pi) - p(s_t|s_{t-1},\pi_0)| \\
&\leq \sum_{s_{t-1}} |p(s_{t-1}|\pi) - p(s_{t-1}|\pi_0)| + \sum_{s_{t-1}} p(s_{t-1}|\pi_0) \sum_{s_t} \sum_{a_{t-1}} p(s_t|s_{t-1},a_{t-1}) |\pi(a_{t-1}|s_{t-1}) - \pi_0(a_{t-1}|s_{t-1})| \\
&= e_{t-1} + \sum_{s_{t-1}} p(s_{t-1}|\pi_0) \sum_{a_{t-1}} |\pi(a_{t-1}|s_{t-1}) - \pi_0(a_{t-1}|s_{t-1})| \\
&= e_{t-1} + \mathbb{E}_{p(s_{t-1}|\pi_0)} \left[ \| \pi(\cdot|s_{t-1}) - \pi_0(\cdot|s_{t-1}) \|_1 \right]
\end{aligned}
\tag{21}
$$

By applying the recurrence term to $e_{t-1}$, we can obtain:

$$
\begin{aligned}
e_t &\leq e_{t-1} + \mathbb{E}_{p(s_{t-1}|\pi_0)} \left[ \| \pi(\cdot|s_{t-1}) - \pi_0(\cdot|s_{t-1}) \|_1 \right] \\
&= \sum_{i < t} \mathbb{E}_{p(s_i|\pi_0)} \left[ \| \pi(\cdot|s_i) - \pi_0(\cdot|s_i) \|_1 \right]
\end{aligned}
\tag{22}
$$

$\square$

**Lemma A.2.** *Under standard regularity assumptions and bounded rewards, there exists a non-negative constant $M$ satisfying the inequality $\left| \sum_{t,s_t} \gamma^t (p(s_t|\pi) - p(s_t|\pi_0)) \sum_{s_{t+1},a_t} p(s_{t+1}, a_t|s_t, \pi) r_t \right| < M \cdot \sum_t \frac{\gamma^t}{1-\gamma} \mathbb{E}_{p(s_t|\pi_0)} \left[ \| \pi(\cdot|s_t) - \pi_0(\cdot|s_t) \|_1 \right]$*

*Proof.* Considering that the sum of probabilities is 1, for any constant $b$, the following equation holds:

$$
b \cdot \sum_{a_t} \pi(a_t|s_t) - \pi_0(a_t|s_t) = 0
\tag{23}
$$

By applying Equation 23 and Lemma A.1, we can complete the proof:

$$\left|\sum_{t,s_t} \gamma^t (p(s_t|\pi) - p(s_t|\pi_0))\sum_{s_{t+1},a_t} p(s_{t+1},a_t|s_t,\pi)r_t\right| = \left|\sum_{t,s_t} \gamma^t (p(s_t|\pi) - p(s_t|\pi_0))(\mathbb{E}_{p(s_{t+1},a_t|s_t,\pi)}[r_t] - b)\right|$$
$$\leq \max_{s_{0:T}} \left|\mathbb{E}_{p(s_{t+1},a_t|s_t,\pi)}[r_t] - b\right| \sum_{t,s_t} \gamma^t |p(s_t|\pi) - p(s_t|\pi_0)|$$
(24)

By setting $b^* = \arg\min_b \max_{s_{0:T}} \left|\mathbb{E}_{p(s_{t+1},a_t|s_t,\pi)}[r_t] - b\right|$ and $M = \max_{s_{0:T}} \left|\mathbb{E}_{p(s_{t+1},a_t|s_t,\pi)}[r_t] - b^*\right|$, the derivation can proceed as follows:

$$\left|\sum_{t,s_t} \gamma^t (p(s_t|\pi) - p(s_t|\pi_0))\sum_{s_{t+1},a_t} p(s_{t+1},a_t|s_t,\pi)r_t\right| \leq \max_{s_{0:T}} \left|\mathbb{E}_{p(s_{t+1},a_t|s_t,\pi)}[r_t] - b\right| \sum_{t,s_t} \gamma^t |p(s_t|\pi) - p(s_t|\pi_0)|$$
$$= M\sum_t \gamma^t e_t$$
$$\leq M\sum_t \gamma^t \sum_{i<t} \mathbb{E}_{p(s_i|\pi_0)}\left[\|\pi(\cdot|s_i) - \pi_0(\cdot|s_i)\|_1\right]$$
$$= M\sum_i \sum_{t>i} \gamma^t \mathbb{E}_{p(s_i|\pi_0)}\left[\|\pi(\cdot|s_i) - \pi_0(\cdot|s_i)\|_1\right]$$
$$\leq M \cdot \sum_t \frac{\gamma^{t+1}}{1-\gamma}\mathbb{E}_{p(s_t|\pi_0)}\left[\|\pi(\cdot|s_t) - \pi_0(\cdot|s_t)\|_1\right]$$
$$< M \cdot \sum_t \frac{\gamma^t}{1-\gamma}\mathbb{E}_{p(s_t|\pi_0)}\left[\|\pi(\cdot|s_t) - \pi_0(\cdot|s_t)\|_1\right]$$
(25)

$\square$

**Lemma A.3.** *When the rewards are bounded, there exists a constant $M_2$ such that inequality* $\left|\sum_{t,s_t} \gamma^{t+1}p(s_t|\pi_0)\sum_{a_t}(\pi_0(a_t|s_t) - \pi(a_t|s_t))\sum_{s_{t+1},a_{t+1}} p(s_{t+1},a_{t+1}|a_t,s_t,\pi_0)Q^{\pi_0}(a_{t+1},s_{t+1})\right| \leq$ $M_2\sum_t \gamma^{t+1}\mathbb{E}_{p(s_t|\pi_0)}\left[\|\pi(\cdot|s_t) - \pi_0(\cdot|s_t)\|_1\right]$ *holds.*

*Proof.* Similar to the proof of Lemma A.2, we can obtain:

$$\left|\sum_{t,s_t} \gamma^{t+1}p(s_t|\pi_0)\sum_{a_t}(\pi_0(a_t|s_t) - \pi(a_t|s_t))\sum_{s_{t+1},a_{t+1}} p(s_{t+1},a_{t+1}|a_t,s_t,\pi_0)Q^{\pi_0}(a_{t+1},s_{t+1})\right|$$
$$= \left|\sum_{t,s_t} \gamma^{t+1}p(s_t|\pi_0)\sum_{a_t}(\pi_0(a_t|s_t) - \pi(a_t|s_t))\left(\mathbb{E}_{p(s_{t+1},a_{t+1}|a_t,s_t,\pi_0)}[Q^{\pi_0}(a_{t+1},s_{t+1})] - b\right)\right|$$
$$\leq \max_{t,s_t,a_t}\left|\mathbb{E}_{p(s_{t+1},a_{t+1}|a_t,s_t,\pi_0)}[Q^{\pi_0}(a_{t+1},s_{t+1})] - b\right|\sum_{t,s_t} \gamma^{t+1}p(s_t|\pi_0)\sum_{a_t}|\pi_0(a_t|s_t) - \pi(a_t|s_t)|$$
(26)

By setting $b^* = \arg\min_b \max_{t,s_t,a_t}\left|\mathbb{E}_{p(s_{t+1},a_{t+1}|a_t,s_t,\pi_0)}[Q^{\pi_0}(a_{t+1},s_{t+1})] - b\right|$ and $M_2 = \max_{t,s_t,a_t}\left|\mathbb{E}_{p(s_{t+1},a_{t+1}|a_t,s_t,\pi_0)}[Q^{\pi_0}(a_{t+1},s_{t+1})] - b^*\right|$, the derivation can proceed as follows:

$$\left|\sum_{t,s_t} \gamma^{t+1}p(s_t|\pi_0)\sum_{a_t}(\pi_0(a_t|s_t) - \pi(a_t|s_t))\sum_{s_{t+1},a_{t+1}} p(s_{t+1},a_{t+1}|a_t,s_t,\pi_0)Q^{\pi_0}(a_{t+1},s_{t+1})\right|$$
$$\leq M_2\sum_{t,s_t} \gamma^{t+1}p(s_t|\pi_0)\sum_{a_t}|\pi_0(a_t|s_t) - \pi(a_t|s_t)|$$
$$= M_2\sum_t \gamma^{t+1}\mathbb{E}_{p(s_t|\pi_0)}\left[\|\pi(\cdot|s_t) - \pi_0(\cdot|s_t)\|_1\right]$$
(27)

$\square$

**Theorem 5.1** Let $\pi_0$ be the current policy and $\pi$ be the updated policy. Under standard regularity assumptions and bounded rewards, the expected discounted reward $\eta(\pi)$ is lower-bounded by the surrogate objective penalized by the L1-norm: $\eta(\pi) \geq L_{\pi_0}(\pi) - C \cdot \sum_t \gamma^t \frac{1+\gamma-\gamma^2}{1-\gamma}\mathbb{E}_{p(s_t|\pi_0)}\left[\|\pi(\cdot|s_t) - \pi_0(\cdot|s_t)\|_1\right]$, where $L_{\pi_0}(\pi)$ is the standard local approximation to $\eta(\pi)$ and $C$ is a constant related to the reward magnitude.

*Proof.* The definition of the general reinforcement learning objective function $\eta$ is:

$$\eta(\pi) = \mathbb{E}_{p(a_{0:T},s_{0:T}|\pi)}\left[\sum_t \gamma^t r_t\right]$$
(28)

where $a_{0:T}$ are the $T$ actions from time 0 to $T$, $s_{0:T}$ is the environment in which these $T$ actions are applied, $r_t$ is the reward obtained after action $a_t$, and $\gamma$ is the discount factor. Now we consider the change in $\eta$ after the policy update from $\pi_0$ to $\pi$:

$$J(\pi) = \eta(\pi) - \eta(\pi_0)$$
$$= \sum_t \gamma^t \sum_{a_t,s_t,s_{t+1}}(p(a_t,s_t,s_{t+1}|\pi) - p(a_t,s_t,s_{t+1}|\pi_0))r_t$$
(29)

Meanwhile, the standard local approximation $L_{\pi_0}(\pi)$ is equivalent to the following objective function:

$$
\begin{aligned}
L_{\pi_0}(\pi) - \eta(\pi_0) &= \sum_{t,a_t,s_t} \gamma^t p(s_t|\pi_0)\pi(a_t|s_t)A^{\pi_0}(a_t,s_t) \\
&= \sum_{t,a_t,s_t} \gamma^t p(s_t|\pi_0)\pi(a_t|s_t)A^{\pi_0}(a_t,s_t) - \sum_{t,a_t,s_t} \gamma^t p(s_t|\pi_0)\pi_0(a_t|s_t)A^{\pi_0}(a_t,s_t) \\
&= \sum_{t,a_t,s_t} \gamma^t p(s_t|\pi_0)A^{\pi_0}(a_t,s_t)\left(\pi(a_t|s_t) - \pi_0(a_t|s_t)\right)
\end{aligned}
\tag{30}
$$

where the conclusion $E_{\pi_0}\left[A^{\pi_0}(a,s)\right] = E_{\pi_0}\left[Q^{\pi_0}(a,s) - V^{\pi_0}(s)\right] = E_{\pi_0}\left[Q^{\pi_0}(a,s)\right] - E_{\pi_0}\left[Q^{\pi_0}(a,s)\right] = 0$ is utilized. Since $V^{\pi_0}(s)\sum_a \pi(a|s) - \pi_0(a|s) = 0$, Equation 30 is equivalent to

$$
L_{\pi_0}(\pi) - \eta(\pi_0) = \sum_{t,a_t,s_t} \gamma^t p(s_t|\pi_0)Q^{\pi_0}(a_t,s_t)\left(\pi(a_t|s_t) - \pi_0(a_t|s_t)\right).
\tag{31}
$$

Now we can consider the error between Equation 31 and Equation 29. Their difference is:

$$
\begin{aligned}
\eta(\pi) - L_{\pi_0}(\pi) &= J(\pi) - (L_{\pi_0}(\pi) - \eta(\pi_0)) \\
&= \sum_t \gamma^t \sum_{a_t,s_t,s_{t+1}} \left(p(a_t,s_t,s_{t+1}|\pi) - p(a_t,s_t,s_{t+1}|\pi_0)\right)r_t \\
&\quad - \sum_{t,a_t,s_t} \gamma^t p(s_t|\pi_0)Q^{\pi_0}(a_t,s_t)\left(\pi(a_t|s_t) - \pi_0(a_t|s_t)\right) \\
&= \sum_t \gamma^t \sum_{a_t,s_t} \left(p(a_t,s_t|\pi) - p(a_t,s_t|\pi_0)\right)\sum_{s_{t+1}} p(s_{t+1}|a_t,s_t)r_t \\
&\quad - \sum_{t,a_t,s_t} \gamma^t p(s_t|\pi_0)Q^{\pi_0}(a_t,s_t)\left(\pi(a_t|s_t) - \pi_0(a_t|s_t)\right) \\
&= \sum_t \gamma^t \sum_{a_t,s_t} p(a_t,s_t|\pi)\sum_{s_{t+1}} p(s_{t+1}|a_t,s_t)r_t - \sum_t \gamma^t \sum_{a_t,s_t} p(a_t,s_t|\pi_0)\sum_{s_{t+1}} p(s_{t+1}|a_t,s_t)r_t \\
&\quad - \sum_t \gamma^t \sum_{a_t,s_t} p(s_t|\pi_0)Q^{\pi_0}(a_t,s_t)\left(\pi(a_t|s_t) - \pi_0(a_t|s_t)\right)
\end{aligned}
\tag{32}
$$

By using the Bellman equation, the second term can be split:

$$
\begin{aligned}
&- \sum_t \gamma^t \sum_{a_t,s_t} p(a_t,s_t|\pi_0)\sum_{s_{t+1}} p(s_{t+1}|a_t,s_t)r_t \\
&= - \sum_t \gamma^t \sum_{a_t,s_t} p(a_t,s_t|\pi_0)\sum_{s_{t+1}} p(s_{t+1}|a_t,s_t)Q^{\pi_0}(a_t,s_t) \\
&\quad + \sum_t \gamma^{t+1} \sum_{a_t,s_t} p(a_t,s_t|\pi_0)\sum_{s_{t+1},a_{t+1}} p(s_{t+1},a_{t+1}|a_t,s_t,\pi_0)Q^{\pi_0}(a_{t+1},s_{t+1}) \\
&= - \sum_t \gamma^t \sum_{a_t,s_t} p(a_t,s_t|\pi_0)Q^{\pi_0}(a_t,s_t) \\
&\quad + \sum_t \gamma^{t+1} \sum_{a_t,s_t} p(a_t,s_t|\pi_0)\sum_{s_{t+1},a_{t+1}} p(s_{t+1},a_{t+1}|a_t,s_t,\pi_0)Q^{\pi_0}(a_{t+1},s_{t+1})
\end{aligned}
\tag{33}
$$

Substituting Equation 33 into Equation 31 and splitting the last term of Equation 31, we obtain:

$$
\begin{aligned}
&\eta(\pi) - L_{\pi_0}(\pi) \\
&= \sum_t \gamma^t \sum_{a_t,s_t} p(a_t,s_t|\pi)\sum_{s_{t+1}} p(s_{t+1}|a_t,s_t)r_t - \sum_t \gamma^t \sum_{a_t,s_t} p(a_t,s_t|\pi_0)Q^{\pi_0}(a_t,s_t) \\
&\quad + \sum_t \gamma^{t+1} \sum_{a_t,s_t} p(a_t,s_t|\pi_0)\sum_{s_{t+1},a_{t+1}} p(s_{t+1},a_{t+1}|a_t,s_t,\pi_0)Q^{\pi_0}(a_{t+1},s_{t+1}) \\
&\quad - \sum_t \gamma^t \sum_{a_t,s_t} p(s_t|\pi_0)Q^{\pi_0}(a_t,s_t)\pi(a_t|s_t) + \sum_t \gamma^t \sum_{a_t,s_t} p(s_t,a_t|\pi_0)Q^{\pi_0}(a_t,s_t) \\
&= \sum_t \gamma^t \sum_{a_t,s_t} p(a_t,s_t|\pi)\sum_{s_{t+1}} p(s_{t+1}|a_t,s_t)r_t \\
&\quad + \sum_t \gamma^{t+1} \sum_{a_t,s_t} p(a_t,s_t|\pi_0)\sum_{s_{t+1},a_{t+1}} p(s_{t+1},a_{t+1}|a_t,s_t,\pi_0)Q^{\pi_0}(a_{t+1},s_{t+1}) \\
&\quad - \sum_t \gamma^t \sum_{a_t,s_t} p(s_t|\pi_0)Q^{\pi_0}(a_t,s_t)\pi(a_t|s_t)
\end{aligned}
\tag{34}
$$

By applying the Bellman equation to the $r_t$ in the first term, we obtain:

$$
\begin{aligned}
\eta(\pi) - L_{\pi_0}(\pi) &= \sum_t \gamma^t \sum_{a_t,s_t} p(a_t, s_t|\pi) Q^{\pi_0}(a_t, s_t) \\
&- \sum_t \gamma^{t+1} \sum_{a_t,s_t} p(a_t, s_t|\pi) \sum_{s_{t+1},a_{t+1}} p(s_{t+1}, a_{t+1}|a_t, s_t, \pi_0) Q^{\pi_0}(a_{t+1}, s_{t+1}) \\
&+ \sum_t \gamma^{t+1} \sum_{a_t,s_t} p(a_t, s_t|\pi_0) \sum_{s_{t+1},a_{t+1}} p(s_{t+1}, a_{t+1}|a_t, s_t, \pi_0) Q^{\pi_0}(a_{t+1}, s_{t+1}) \\
&- \sum_t \gamma^t \sum_{a_t,s_t} p(s_t|\pi_0) Q^{\pi_0}(a_t, s_t) \pi(a_t|s_t) \\
&= \sum_t \gamma^t \sum_{a_t,s_t} \pi(a_t|s_t)(p(s_t|\pi) - p(s_t|\pi_0)) Q^{\pi_0}(a_t, s_t) \\
&- \sum_t \gamma^{t+1} \sum_{a_t,s_t} \pi(a_t|s_t) p(s_t|\pi) \sum_{s_{t+1},a_{t+1}} p(s_{t+1}, a_{t+1}|a_t, s_t, \pi_0) Q^{\pi_0}(a_{t+1}, s_{t+1}) \\
&+ \sum_t \gamma^{t+1} \sum_{a_t,s_t} \pi_0(a_t|s_t) p(s_t|\pi) \sum_{s_{t+1},a_{t+1}} p(s_{t+1}, a_{t+1}|a_t, s_t, \pi_0) Q^{\pi_0}(a_{t+1}, s_{t+1}) \\
&- \sum_t \gamma^{t+1} \sum_{a_t,s_t} \pi_0(a_t|s_t) p(s_t|\pi) \sum_{s_{t+1},a_{t+1}} p(s_{t+1}, a_{t+1}|a_t, s_t, \pi_0) Q^{\pi_0}(a_{t+1}, s_{t+1}) \\
&+ \sum_t \gamma^{t+1} \sum_{a_t,s_t} p(a_t, s_t|\pi_0) \sum_{s_{t+1},a_{t+1}} p(s_{t+1}, a_{t+1}|a_t, s_t, \pi_0) Q^{\pi_0}(a_{t+1}, s_{t+1})
\end{aligned}
\tag{35}
$$

Merging the adjacent terms once more yields:

$$
\begin{aligned}
&\eta(\pi) - L_{\pi_0}(\pi) \\
&= \sum_t \gamma^t \sum_{a_t,s_t} \pi(a_t|s_t)(p(s_t|\pi) - p(s_t|\pi_0)) Q^{\pi_0}(a_t, s_t) \\
&+ \sum_t \gamma^{t+1} \sum_{a_t,s_t} (\pi_0(a_t|s_t) - \pi(a_t|s_t)) p(s_t|\pi) \sum_{s_{t+1},a_{t+1}} p(s_{t+1}, a_{t+1}|a_t, s_t, \pi_0) Q^{\pi_0}(a_{t+1}, s_{t+1}) \\
&+ \sum_t \gamma^{t+1} \sum_{a_t,s_t} \pi_0(a_t|s_t)(p(s_t|\pi_0) - p(s_t|\pi)) \sum_{s_{t+1},a_{t+1}} p(s_{t+1}, a_{t+1}|a_t, s_t, \pi_0) Q^{\pi_0}(a_{t+1}, s_{t+1}) \\
&= \sum_{t,s_t} \gamma^t (p(s_t|\pi) - p(s_t|\pi_0)) \sum_{a_t} \pi(a_t|s_t) Q^{\pi_0}(a_t, s_t) \\
&+ \sum_{t,s_t} \gamma^{t+1} p(s_t|\pi) \sum_{a_t} (\pi_0(a_t|s_t) - \pi(a_t|s_t)) \sum_{s_{t+1},a_{t+1}} p(s_{t+1}, a_{t+1}|a_t, s_t, \pi_0) Q^{\pi_0}(a_{t+1}, s_{t+1}) \\
&- \sum_{t,s_t} \gamma^{t+1} (p(s_t|\pi_0) - p(s_t|\pi)) \sum_{a_t} \pi(a_t|s_t) \sum_{s_{t+1},a_{t+1}} p(s_{t+1}, a_{t+1}|a_t, s_t, \pi_0) Q^{\pi_0}(a_{t+1}, s_{t+1}) \\
&+ \sum_{t,s_t} \gamma^{t+1} (p(s_t|\pi_0) - p(s_t|\pi)) \sum_{a_t} \pi(a_t|s_t) \sum_{s_{t+1},a_{t+1}} p(s_{t+1}, a_{t+1}|a_t, s_t, \pi_0) Q^{\pi_0}(a_{t+1}, s_{t+1}) \\
&+ \sum_{t,s_t} \gamma^{t+1} (p(s_t|\pi_0) - p(s_t|\pi)) \sum_{a_t} \pi_0(a_t|s_t) \sum_{s_{t+1},a_{t+1}} p(s_{t+1}, a_{t+1}|a_t, s_t, \pi_0) Q^{\pi_0}(a_{t+1}, s_{t+1})
\end{aligned}
\tag{36}
$$

Merging the adjacent terms yields and applying the Bellman equation:

$$
\begin{aligned}
&\eta(\pi) - L_{\pi_0}(\pi) \\
&= \sum_{t,s_t} \gamma^t (p(s_t|\pi) - p(s_t|\pi_0)) \sum_{a_t} \pi(a_t|s_t) \left( Q^{\pi_0}(a_t, s_t) - \gamma \sum_{s_{t+1},a_{t+1}} p(s_{t+1}, a_{t+1}|a_t, s_t, \pi_0) Q^{\pi_0}(a_{t+1}, s_{t+1}) \right) \\
&+ \sum_{t,s_t} \gamma^{t+1} p(s_t|\pi) \sum_{a_t} (\pi_0(a_t|s_t) - \pi(a_t|s_t)) \sum_{s_{t+1},a_{t+1}} p(s_{t+1}, a_{t+1}|a_t, s_t, \pi_0) Q^{\pi_0}(a_{t+1}, s_{t+1}) \\
&+ \sum_{t,s_t} \gamma^{t+1} (p(s_t|\pi) - p(s_t|\pi_0)) \sum_{a_t} (\pi(a_t|s_t) - \pi_0(a_t|s_t)) \sum_{s_{t+1},a_{t+1}} p(s_{t+1}, a_{t+1}|a_t, s_t, \pi_0) Q^{\pi_0}(a_{t+1}, s_{t+1}) \\
&= \sum_{t,s_t} \gamma^t (p(s_t|\pi) - p(s_t|\pi_0)) \sum_{a_t,s_{t+1}} p(a_t, s_{t+1}|s_t, \pi) r_t \\
&+ \sum_{t,s_t} \gamma^{t+1} p(s_t|\pi_0) \sum_{a_t} (\pi_0(a_t|s_t) - \pi(a_t|s_t)) \sum_{s_{t+1},a_{t+1}} p(s_{t+1}, a_{t+1}|a_t, s_t, \pi_0) Q^{\pi_0}(a_{t+1}, s_{t+1})
\end{aligned}
\tag{37}
$$

By applying the triangle inequality and Lemma A.2, A.3, we obtain:

$$
\begin{aligned}
&|\eta(\pi) - L_{\pi_0}(\pi)| \\
&\leq \left| \sum_{t,s_t} \gamma^t (p(s_t|\pi) - p(s_t|\pi_0)) \sum_{a_t,s_{t+1}} p(a_t, s_{t+1}|s_t, \pi) r_t \right| \\
&+ \left| \sum_{t,s_t} \gamma^{t+1} p(s_t|\pi_0) \sum_{a_t} (\pi_0(a_t|s_t) - \pi(a_t|s_t)) \sum_{s_{t+1},a_{t+1}} p(s_{t+1}, a_{t+1}|a_t, s_t, \pi_0) Q^{\pi_0}(a_{t+1}, s_{t+1}) \right| \\
&< M \cdot \sum_t \frac{\gamma^t}{1-\gamma} \mathbb{E}_{p(s_t|\pi_0)} \left[ \|\pi(\cdot|s_t) - \pi_0(\cdot|s_t)\|_1 \right] + M_2 \sum_t \gamma^{t+1} \mathbb{E}_{p(s_t|\pi_0)} \left[ \|\pi(\cdot|s_t) - \pi_0(\cdot|s_t)\|_1 \right]
\end{aligned}
\tag{38}
$$

By setting $C = M + M_2$, the proof can be completed. $\qquad\square$

**Theorem 5.2** Assume rewards are bounded, $\pi_{\text{ref}}$ does not approach $\pi^*$, and $\beta > 0$. Let $\pi^*$ denote the policy that maximizes the expected reward and $a^*$ be the action with the highest reward. In scenarios where $\pi_{\text{ref}}(a^*|s) \to 0$ or $\exp(\frac{r(a^*|s)}{\beta}) \to \mathbb{E}_{\pi_{\text{ref}}} \left[ \exp(\frac{r(a,s)}{\beta}) \right]$, it holds that $k(\pi^*_{KL}) \to 1$, whereas $k(\pi^*_{L_1})$ remains bounded away from 1. Crucially, in the specific case where $\pi_{\text{ref}}(a^*|s) \to 0$ and $d = \mathbb{D}_{KL}$, we achieve $k(\pi^*_{L_1}) \to 0$.

*Proof.* For convenience, we will omit the state $s$ in the following derivations. Obviously, the optimal policy is:

$$\pi^*(a) = \delta(a - a^*) = \delta_{a^*}(a) \tag{39}$$

Therefore, for any policy $\pi$, its KL divergence is:

$$\mathbb{D}_{KL}(\pi^* || \pi) = \sum_a \pi^*(a) \log \frac{\pi^*(a)}{\pi(a)} = -\log \pi(a^*) \tag{40}$$

Similarly, the L1-norm is:

$$\|\pi - \pi^*\|_1 = \sum_a |\pi(a) - \pi^*(a)| = 1 - \pi(a^*) + \sum_{a \neq a^*} \pi(a) = 1 - \pi(a^*) + (1 - \pi(a^*)) = 2(1 - \pi(a^*)) \tag{41}$$

Substituting the optimal solutions for L1 training (Equation 5) and KL training (Equation 4) into Equation 40 and Equation 41, respectively, and then calculating the improvement value $k$, we obtain:

$$k(\pi_{KL}^*, D_{KL}) = \frac{-\log \pi_{\text{ref}}(a^*) - \log \frac{\exp(\frac{r(a^*)}{\beta})}{E_{\pi_{\text{ref}}}[\exp(\frac{r(a)}{\beta})]}}{-\log \pi_{\text{ref}}(a^*)} \tag{42}$$

$$k(\pi_{L_1}^*, D_{KL}) = \frac{-\log \pi_{\text{ref}}(a^*) + \log \frac{\pi_{\text{ref}}(a^*)}{\pi_{\text{ref}}(a^*) + \Delta_{a,s}^+}}{-\log \pi_{\text{ref}}(a^*)} \tag{43}$$

$$k(\pi_{KL}^*, \|\cdot\|_1) = \frac{1 - \frac{\exp(\frac{r(a^*)}{\beta})}{E_{\pi_{\text{ref}}}[\exp(\frac{r(a)}{\beta})]} \pi_{\text{ref}}(a^*)}{1 - \pi_{\text{ref}}(a^*)} \tag{44}$$

$$k(\pi_{L_1}^*, \|\cdot\|_1) = \frac{1 - \pi_{\text{ref}}(a^*) - \Delta_{a,s}^+}{1 - \pi_{\text{ref}}(a^*)} \tag{45}$$

Substituting the corresponding conditions into the respective equations completes the proof.

$\square$

### A.3. Extension to Multi-Step MDPs

The statements and proofs of Theorem 4.1 and Theorem 5.2 are presented based on a response-level, bandit-style formulation for clarity. This formulation is standard in the analysis of RLHF-style methods and can be naturally extended to token-level optimization by interpreting the reward $r$ in our derivation as the optimal action-value function $Q^*$ under the optimal policy $\pi^*$. This is because the generalized multi-step formulation is equivalent to the bandit-style optimization problem:

$$\pi^* = \arg\max_\pi \left( \mathbb{E}_\pi[Q^*(s, a)] - \Omega(\pi) \right), \tag{46}$$

where $\Omega$ denotes the regularizer (Geist et al., 2019). Importantly, our theorems are intended to characterize the properties of the optimal policy under L1 regularization, rather than to explicitly compute $Q^*$ in practice. Therefore, this formulation allows the theoretical properties to carry over to the token-level updates used in GRPO-style training without compromising practical applicability.

### A.4. Entropy Analysis

Although the sparsity-inducing property of the L1-norm intuitively tends to reduce the entropy of the model, which is considered undesirable during the RLVR stage, in practice, the L1 constraint, like other constraints, also mitigates the decrease in policy entropy. As shown in Cui et al. (2025), entropy change under small updates is proportional to the covariance between the log-policy and the policy gradient. Since both the KL-divergence and L1-norm provide gradients negatively correlated with the policy ($-\log \frac{\pi(a|s)}{\pi_{\text{ref}}(a|s)}$ and $-\text{sgn}(\pi(a|s) - \pi_{\text{ref}}(a|s))$, respectively), both inherently mitigate entropy collapse. From a mechanistic perspective, the sparsity induced by L1 regularization tends to preserve the probabilities of medium-reward actions close to the reference policy, which helps avoid entropy collapse.

## A.5. Offline MRPO

The objective functions of both MRPO-G and MRPO-R satisfy the following form:

$$\hat{\mathcal{L}} = \sum_s \mathcal{L}_1(s) - \beta_s \mathcal{L}_2(s) \tag{47}$$

where we denote $\beta_s$ as Lagrange multiplier for state $s$. The derivative with respect to $\beta_s$ is:

$$\frac{d\hat{\mathcal{L}}}{d\beta_s} = -\mathcal{L}_2(s) \tag{48}$$

We initialize $\beta_s$ with $\beta$, and then perform one gradient ascent step on it using a learning rate $\alpha\beta$:

$$\beta_s = \beta(1 + \alpha\mathcal{L}_2(s)) \tag{49}$$

Rescaling the objective function yields:

$$\frac{\beta}{\beta_s}\hat{\mathcal{L}} = \sum_s \lambda_s \mathcal{L}_1(s) - \beta\mathcal{L}_2(s) \tag{50}$$

This finishes the proof.

## A.6. More Properties of the L1-Norm

### A.6.1. SERVING AS A LOWER BOUND FOR THE KL DIVERGENCE IN CERTAIN CASES

We first present several lemmas, and then use these lemmas to proceed with the proof.

**Lemma A.4.** *Let $y$ be a real number satisfying $y > 0$, then $\frac{1+y}{y}\log(1+y) \geq 1$ and $\log(1+y) \geq \frac{y}{1+y}$ holds.*

*Proof.* When $x > 0$, the following inequality holds:

$$\log(x) \leq x - 1 \tag{51}$$

Let $x = \frac{1}{1+y}$, then Equation 51 can be transformed into:

$$\log(1+y) \geq \frac{y}{1+y} \tag{52}$$

When $y > 0$, Equation 52 is equivalent to:

$$\frac{1+y}{y}\log(1+y) \geq 1 \tag{53}$$

$\square$

**Lemma A.5.** *Let $z$ be a real number satisfying $z > 0$, then $z - \log(1+z) \geq 0$ and $1 + \log(1+z) \geq \frac{(1+z)\log(1+z)}{z}$ holds.*

*Proof.* We further set $x = 1 + z$ and substitute it into Equation 51, obtaining:

$$z - \log(1+z) \geq 0 \tag{54}$$

Where the left-hand side of the inequality is equivalent to:

$$z - \log(1+z) = z + z\log(1+z) - (1+z)\log(1+z) \tag{55}$$

Therefore, when $z > 0$, the following inequality holds:

$$1 + \log(1+z) - \frac{(1+z)\log(1+z)}{z} = \frac{1}{z}(z + z\log(1+z) - (1+z)\log(1+z)) \geq 0 \tag{56}$$

$\square$

**Lemma A.6.** *Let $k$ and $z$ be two real numbers satisfying $0 < z + 1 \leq k$. Then $k \log(k) \geq \frac{(1+z)\log(1+z)}{z}(k-1)$ holds.*

*Proof.* We will finish the proof by considering the function $f$:

$$f(k) = k \log(k) - \frac{(1+z)\log(1+z)}{z}(k-1) \tag{57}$$

Clearly, $f(1+z) = 0$. We now wish to examine the case where $k \geq 1 + z$. Therefore, we take the derivative of $f$:

$$\frac{\mathrm{d}}{\mathrm{d}k}f(k) = 1 + \log(k) - \frac{(1+z)\log(1+z)}{z} \geq 1 + \log(1+z) - \frac{(1+z)\log(1+z)}{z} \geq 0 \tag{58}$$

Therefore, we can conclude that when $k \geq 1 + z$, $f(k) \geq 0$, which can also be expressed as:

$$k \log(k) \geq \frac{(1+z)\log(1+z)}{z}(k-1) \tag{59}$$

$\square$

**Lemma A.7.** *Let $p$ and $q$ be two distributions subject to $1 + \epsilon = \inf_{p(x)>q(x)} \frac{p(x)}{q(x)}$. Then, $\sum_{p(x)>q(x)} p(x) \log\left(\frac{p(x)}{q(x)}\right) \geq \frac{(1+\epsilon)\log(1+\epsilon)}{\epsilon}\sum_{p(x)>q(x)} p(x) - q(x)$ holds.*

*Proof.* Since the sum of probabilities is 1, we have:

$$
\begin{aligned}
\sum_{p(x)>q(x)} p(x) - q(x) &= \sum_x p(x) - q(x) - \sum_{p(x)\leq q(x)} p(x) - q(x) \\
&= \sum_x p(x) - \sum_x q(x) - \sum_{p(x)\leq q(x)} p(x) - q(x) \\
&= 1 - 1 - \sum_{p(x)\leq q(x)} p(x) - q(x) \\
&= -\sum_{p(x)\leq q(x)} p(x) - q(x)
\end{aligned}
\tag{60}
$$

Therefore, without loss of generality, let $u = \sum_{p(x)>q(x)} p(x) - q(x)$. Then the L1 norm is:

$$\|p - q\|_1 = \sum_x |p(x) - q(x)| = \sum_{p(x)>q(x)} p(x) - q(x) + \sum_{q(x)\geq p(x)} q(x) - p(x) = 2u \tag{61}$$

We assume $1 + \epsilon = \inf_{p(x)>q(x)} \frac{p(x)}{q(x)}$, then we have:

$$p(x) \geq (1+\epsilon)q(x), \quad \forall p(x) > q(x) \tag{62}$$

Now, we further consider the minimum value of the following function:

$$
\begin{aligned}
g &= \max_{\substack{\sum_{p(x)>q(x)} p(x) - q(x) = u, \\ p \text{ and } q \text{ are probabilities}}} \sum_{p(x)>q(x)} p(x) \log\left(\frac{p(x)}{q(x)}\right) \\
&\geq \max_{\sum_{p(x)>q(x)} p(x) - q(x) = u} \sum_{p(x)>q(x)} p(x) \log\left(\frac{p(x)}{q(x)}\right) = \hat{g}
\end{aligned}
\tag{63}
$$

By constructing the Lagrangian function to $\hat{g}$:

$$\mathcal{L}(\hat{g}, \lambda) = \sum_{p(x)>q(x)} p(x) \log\left(\frac{p(x)}{q(x)}\right) + \lambda\left(\sum_{p(x)>q(x)} p(x) - q(x) - u\right) \tag{64}$$

We find its stationary points:

$$1 + \log(p(x)) - \log(q(x)) + \lambda = 0 \tag{65}$$

We know that when $\hat{g}$ is minimized, the relationship between $p$ and $q$ is:

$$p(x) = e^{-1-\lambda}q(x) = kq(x) \tag{66}$$

Therefore, $g$ satisfies the inequality:

$$g \geq \hat{g} = \sum_{p(x)>q(x)} kq(x) \log\left(\frac{kq(x)}{q(x)}\right) = k \log(k) \sum_{p(x)>q(x)} q(x) \tag{67}$$

At this point, $u$ satisfies the relation:

$$u = \sum_{p(x)>q(x)} p(x) - q(x) = (k-1) \sum_{p(x)>q(x)} q(x) \tag{68}$$

Therefore, substituting Equations 67 and 62, 68 and 59, the following relation can be obtained:

$$
\begin{aligned}
&\sum_{p(x)>q(x)} p(x) \log\left(\frac{p(x)}{q(x)}\right) - \frac{(1+\epsilon)\log(1+\epsilon)}{\epsilon} \sum_{p(x)>q(x)} p(x) - q(x) \\
&\geq k\log(k) \sum_{p(x)>q(x)} q(x) - \frac{(1+\epsilon)\log(1+\epsilon)}{\epsilon} u \\
&= \left(k\log(k) - \frac{(1+\epsilon)\log(1+\epsilon)}{\epsilon}(k-1)\right) \sum_{p(x)>q(x)} q(x) \\
&\geq 0
\end{aligned}
\tag{69}
$$

$\square$

**Theorem A.8.** *If $p$ and $q$ are two probability distributions satisfying $\min_{p(x)>q(x)} \frac{p(x)}{q(x)} = 1 + \epsilon$, then there always exists $\tau_3 > 0$ such that $\|p - q\|_1 \leq \tau_3 D_{KL}(p\|q)$*

*Proof.* Using Equation 69, a lower bound for the KL divergence can be derived:

$$
\begin{aligned}
D_{KL}(p\|q) &= \sum_x p(x) \log\left(\frac{p(x)}{q(x)}\right) \\
&= \sum_{p(x)>q(x)} p(x) \log\left(\frac{p(x)}{q(x)}\right) + \sum_{p(x)\leq q(x)} p(x) \log\left(\frac{p(x)}{q(x)}\right) \\
&\geq \frac{(1+\epsilon)\log(1+\epsilon)}{\epsilon} \sum_{p(x)>q(x)} p(x) - q(x) - \sum_{p(x)\leq q(x)} p(x) \log\left(\frac{q(x)}{p(x)}\right) \\
&\geq \frac{(1+\epsilon)\log(1+\epsilon)}{\epsilon} \sum_{p(x)>q(x)} p(x) - q(x) - \sum_{p(x)\leq q(x)} p(x) \left(\frac{q(x)}{p(x)} - 1\right) \\
&= \frac{(1+\epsilon)\log(1+\epsilon)}{\epsilon} \sum_{p(x)>q(x)} p(x) - q(x) - \sum_{p(x)\leq q(x)} q(x) - p(x)
\end{aligned}
\tag{70}
$$

Substituting into Equation 60 again, we obtain:

$$
\begin{aligned}
&\frac{(1+\epsilon)\log(1+\epsilon)}{\epsilon} \sum_{p(x)>q(x)} p(x) - q(x) - \sum_{p(x)\leq q(x)} p(x) - q(x) \\
&= \frac{(1+\epsilon)\log(1+\epsilon)}{\epsilon} \sum_{p(x)>q(x)} p(x) - q(x) - \sum_{p(x)>q(x)} p(x) - q(x) \\
&= \left(\frac{(1+\epsilon)\log(1+\epsilon)}{\epsilon} - 1\right) \sum_{p(x)>q(x)} p(x) - q(x) \\
&= \frac{1}{2}\left(\frac{(1+\epsilon)\log(1+\epsilon)}{\epsilon} - 1\right) \|p - q\|_1
\end{aligned}
\tag{71}
$$

Therefore, setting $\tau_3^{-1} = \frac{1}{2}\left(\frac{(1+\epsilon)\log(1+\epsilon)}{\epsilon} - 1\right)$ and noting that from Equation 53, we have $\tau_3 > 0$, the proof is completed.

$\square$

# B. Experiment Settings

## B.1. RLHF

To maintain fairness across all baseline methods, we ensured that hyperparameters were kept identical or as closely aligned as possible. Each experiment was assigned 80GB A100 GPU resources, and all runs relied on trl version 0.19.1 along with

*Table 6.* Details on hyperparameter in RLHF experiments.

| Hyperparameter | value |
|---|---|
| learning rate | 1e-5 |
| epoch | 1 |
| num generations | 3 |
| gradient accumulation steps | 32 |
| per device train batch size | 3 |
| generation batch size | 30 |
| bf16 | True |
| bf16 full eval | True |
| LoRA r | 8 |
| LoRA alpha | 16 |
| max completion length | 512 |

*Table 7.* AlpacaEval performance under different GRPO beta values on Meta-Llama3.1-8B-Instruct.

| beta | AlpacaEval-WR | AlpacaEval-LC |
|---|---|---|
| Base Model | 42.13 | 46.72 |
| 0.1 | 50.98 | 57.22 |
| 0.02 | 53.49 | 57.96 |
| 0.01 | **53.97** | **60.51** |
| 0.005 | 43.25 | 50.18 |
| 0.001 | 45.05 | 49.82 |

the matching dependency versions. PPO was the only method that required two A100 GPUs, as it trains the critic and policy models simultaneously; all other methods used a single A100 GPU. We applied the same LoRA fine-tuning setup to every LLM to reduce memory usage, choosing a rank of 8 and an alpha value of 16. The maximum response length was capped at 512 tokens, slightly above the typical output length of 7B models to ensure that truncation would rarely occur. For models with at least 7B parameters, we used a group size of 3; for smaller models, the group size was set to 4. All models were trained with a batch size of 96. Following prior work, each experiment was run for one epoch, except for PPO, which effectively uses a group size of 1 and therefore was trained for 3 epochs. Under these settings, training a 7B-scale model or larger takes around 120 hours, a 3B model takes roughly 80 hours, and a 1.5B model finishes in about 45 hours. More details can be found in Table 6.

**PPO.** We configured PPO with a beta value of 0.01, the same value used in GRPO, and set gamma to 0.95 for the advantage computation. The critic was initialized from the reward model according to the official trl example and further fine-tuned using LoRA with rank 4 and alpha 8 to avoid out-of-memory issues. All other hyperparameters, such as learning rate and the loss weights for policy and critic updates, remained at their default trl settings.

**GRPO, BNPO, and Dr-GRPO.** To determine the best beta value for GRPO, we ran experiments with beta set to 0.1, 0.02, 0.01, 0.005, and 0.001. Based on performance, we selected 0.01 as the final choice. The results for the different beta values appear in Table 7. Because BNPO and Dr-GRPO share the same underlying objective structure, differing only in how the loss is averaged, and both rely on KL-based constraints, they also use the same beta value as GRPO.

**DPO and IPO.** Our online DPO and online IPO implementations used trl's OnlineDPOTrainer. The beta value was fixed at 0.01 for consistency with GRPO. Since using the default learning rate of 1e-6 yielded only modest improvements, we increased the rate to 3e-6. All other hyperparameters, including label smoothing, followed trl's default configuration.

**DPH-F, DPH-JS, f-PO.** These methods were implemented exactly as described in their original papers, using GRPOTrainer as the base class in trl and overriding the $\_compute\_loss$ function. For f-PO, we adopted the list-wise objective introduced in the paper. DPH-F and DPH-JS (Li et al., 2025) used the same learning rate as GRPO, while f-PO used its best-performing value, namely 1e-5 and for f-PO. All three methods used a beta value of 0.01 to ensure comparability with GRPO.

**MRPO-G and MRPO-R.** Both MRPO variants were implemented by subclassing GRPOTrainer in trl and replacing the $\_compute\_loss$ method. Given that the gradient magnitude of the L1 norm is typically about ten times larger than that of the

*Table 8.* Details on hyperparameter in RLVR experiments.

| Hyperparameter | GSM8k | Code-R1 |
|---|---|---|
| learning rate | 3e-6 | 1e-6 |
| step | 11,200 | 22,000 |
| generation batch size | 8 | 4 |
| group size | 8 | 16 |
| batch size per gpu | 1 | 1 |
| generation update step | 64 | 128 |
| gradient accumulation | 64 | 64 |
| max tokens | 1280 | 4096 |

KL divergence, we set the beta for MRPO to 0.001, and for models smaller than 7B, we set it to 0.002 to prevent overfitting. All other hyperparameters, including the clipping threshold, are kept identical to those in GRPO.

**Evaluations.** All evaluations used official implementations. For AlpacaEval, we reported Version 2's Win Rate and Length-Controlled Win Rate, as well as Version 1's Win Rate. Response generation for all models employed a repetition penalty of 1.2, a temperature of 0.7, and a top_p value of 0.95. For Arena-Hard, we used the win rate metrics from both v0.1 and v2.0.

## B.2. RLVR Experiment

**Codes.** Our implementation builds entirely on the KW-R1 framework (Liang et al., 2025), a reinforcement learning codebase with around 1.5k stars on GitHub.

**Prompts.** To ensure the model explicitly produces a chain of thought in a standardized format, we adopted a system prompt closely aligned with the one used in DeepSeek-R1 (Guo et al., 2025) for math-related tasks:

```
You are a helpful assistant. A conversation between User and Assistant. The user \
asks a question, and the Assistant solves it. The Assistant first thinks about the \
reasoning process in the mind and then provides the user with the answer.
The reasoning process and answer are enclosed within <think> </think> and \
<answer> </answer> tags, respectively, i.e., <think> reasoning process here \
</think> <answer> answer here </answer>.
```

For coding tasks, we designed a similar instruction to encourage structured reasoning:

```
You are a helpful programming assistant. The user will ask you a question and \
you as the assistant solve it. The assistant first thinks how to solve the task\
through reasoning and then provides the user with the final answer. The \
reasoning process and answer are enclosed within <think>...</think> and \
<answer>...</answer> tags, respectively.
```

User queries were kept exactly as they appear in the dataset.

**Rewards.** The reward configuration for math tasks mirrors that of KW-R1 (Liang et al., 2025). A correct final answer earns 1 point, and producing the correct output format yields another 0.2 points; all other cases receive 0. The total reward is the combined value of these two components. For coding tasks, we adopted the reward function from Code-R1 (Liu & Zhang, 2025). Any response with incorrect format receives -1.1, regardless of correctness. If the format is correct but the output is wrong, the reward is 0. Only when both the format and result are correct does the model receive a reward of 1.

**Training.** All experiments were run on 8 H100 GPUs. For math tasks, we allocated 1 GPU for rollout generation, 1 GPU for computing the reference policy, and 6 GPUs for training. For code tasks, rollout used 2 GPUs, the reference policy used 1 GPU, and training occupied the remaining 5 GPUs. To check code execution safely, we also launched 32 isolated processes. Because GSM8k is relatively simple, we set its group size to 8, while the more complex code tasks used a group size of 16. The LLM was fine-tuned with full parameters. The beta value for all baselines was set to 0.001, which we found to be the best-performing value for GRPO. GSM8k training ran for 11,200 steps (about 8 hours), whereas Code-R1 training required

*Table 9.* Details on hyperparameter in offline experiments.

| Hyperparameter | value |
|---|---|
| learning rate | 1e-6 |
| epoch | 1 |
| gradient accumulation steps | 64 |
| per device train batch size | 1 |
| bf16 | True |
| bf16 full eval | True |
| LoRA r | 8 |
| LoRA alpha | 16 |
| max completion length | 784 |
| label smoothing | 0 |
| precompute ref log probs | False |
| force use ref model | True |

*Table 10.* Results on TruthfulQA using Qwen2.5-1.5B-Instruct.

| GRPO | MRPO-G | MRPO-R |
|---|---|---|
| 24.36 | 37.09 | 44.06 |

22,000 steps (around 20 hours). The learning rates were 3e-6 for GSM8k and 1e-6 for Code-R1, both validated as optimal under GRPO. More details can be found in Table 8.

**Evaluation.** Final evaluations were conducted on the test sets of GSM8k and Code-R1. For GSM8k, we computed accuracy solely based on the correctness of the answer, ignoring formatting. For Code-R1, both correctness and format were taken into account. The final evaluation metric was the probability that the model produces at least one correct solution within 1, 3, and 5 attempts.

### B.3. Offline RL Experiment

**Training parameters.** All baseline models (e.g., SimPO (Meng et al., 2024), RPO (Pang et al., 2024), SLiC (Zhao et al., 2023), IPO (Azar et al., 2024)) were trained using trl, and through validation on DPO we confirmed that the optimal beta value is 0.05. Each model was trained for a single epoch, which required roughly 9 hours on one H100 GPU. We capped the maximum response length at 784 tokens and applied LoRA to all experiments, using a rank of 8 and an alpha of 16. For MRPO, the implementation was built by extending the DPOTrainer class and modifying only the loss function. In addition, the $\alpha$ for MRPO was set to 0.5. More details can be found in Table 9.

**Evaluation parameters.** Evaluations followed the default settings provided by the corresponding libraries. This included using their built-in sampling temperatures, prompt formats, and the default number of few-shot examples.

## C. More Experimental Results

### C.1. Language Focused Benchmark

Our current evaluation already includes two widely used language-focused benchmarks, AlpacaEval and Arena-Hard. In addition, we have obtained TruthfulQA (Lin et al., 2022) results on Qwen2.5-1.5B-Instruct using EvalScope, as shown in Table 10, which are consistent with our experiments. More language benchmarks will be included in future work.

