# OpenReview forum: "MRPO: Magnitude-Regularized Policy Optimization via L1 Constraints"
_ICML.cc/2026/Conference — ICML 2026 regular_

### Official Review · Reviewer_yLxY · 2026-03-10

**Soundness:** 3
**Presentation:** 3
**Significance:** 3
**Originality:** 3
**Overall Recommendation:** 4
**Confidence:** 4

**Summary:**

The authors propose MRPO, a new reinforcement learning algorithm for LLMs that replaces the KL-divergence constraint with an L1-norm constraint. Unlike KL divergence, which depends on the reference model’s probability distribution, the L1 constraint allows larger updates even for low-probability actions. The authors demonstrate the effectiveness of MRPO through a variety of experiments.

**Compliance With Llm Reviewing Policy:**

Affirmed.

**Final Justification:**

This paper is experimentally sound and makes a significant contribution. I am leaning toward accepting it.

**Key Questions For Authors:**

### Questions
- Since the L1 constraint tends to reduce entropy and may push the policy distribution aggressively toward certain actions relative to the reference model, I am curious about the authors’ perspective on this issue.
- What happens if the constraint with respect to the reference model is completely removed? Do the authors observe instability or performance degradation in that case?

**Limitations:**

yes

**Strengths And Weaknesses:**

### Strengths
----
- **Clear motivation and novelty**
  Replacing the KL-divergence term with an L1 constraint is an interesting idea with clear motivation. The use and limitations of the KL-divergence term have been an active topic of interest in the community, and this work contributes to that discussion.

- **Well-written and clearly presented**
  The paper is clearly written and easy to follow. The experimental results generally support the authors’ claims.
----
### Weaknesses
----
- **Lack of analysis regarding entropy**
  In reinforcement learning, KL-divergence is often used not only for stability but also because it implicitly helps maintain a certain level of policy entropy by preventing the policy from drifting too far from the reference model. In contrast, L1 regularization has sparsity-inducing properties, which may encourage the policy distribution to concentrate excessively on certain actions, potentially leading to policy collapse. Therefore, it would be helpful to include additional theoretical or empirical analysis on how MRPO affects policy entropy and whether it introduces any risk of policy collapse. While maximizing entropy may not be an explicit objective in LLM RL, which already differs from conventional RL in several aspects, such analysis would still strengthen the paper.

---

> ### Author Rebuttal · Authors · 2026-03-30
>
> ### We appreciate the reviewer’s constructive feedback and provide the following analysis regarding policy entropy.
>
> ### Weakness
> We thank the reviewer for raising this important point regarding policy entropy and potential collapse. We agree that maintaining sufficient entropy to preserve exploration capability is important for RL training. However, L1 regularization does not inherently lead to lower entropy or policy collapse. As shown in [1], entropy change under small updates is proportional to the covariance between the log-policy and the policy gradient. Since both the KL-divergence and L1-norm provide gradients negatively correlated with the policy ($ -\log\frac{\pi}{\pi_{ref}} $ and $ - \text{sgn}(\pi - \pi_{ref}) $, respectively), both inherently mitigate entropy collapse. From a mechanistic perspective, the sparsity induced by L1 regularization tends to preserve the probabilities of medium-reward actions close to the reference policy, which helps avoid entropy collapse. Empirically, we do not observe entropy collapse in our experiments. On the contrary, as shown in Figure 3, MRPO produces a broader reward distribution, indicating more effective exploration across actions with diverse reward levels. We will include additional analysis of entropy dynamics in the revision to further clarify this behavior.
>
> ### Q1
> As discussed above, L1 regularization does not inherently reduce entropy. Although L1 may push the policy distribution more aggressively, this effect is selective rather than global: due to its greedy nature, MRPO affects the fewest possible actions when transferring the same amount of probability mass. As a result, only actions with extremely low or extremely high rewards are significantly affected, while the probabilities of medium-reward actions remain unchanged. This, to some extent, helps maintain entropy stability.
>
> ### Q2
> Removing the reference constraint generally leads to instability or performance degradation. In RLHF settings, the reward model is often imperfect, and without a constraint, the policy tends to exploit reward model biases, resulting in policy collapse. As shown in Table 6, decreasing the constraint coefficient significantly degrades performance. In RLVR settings, although rewards are more accurate, removing the constraint does not cause a sharp drop in performance, but it still leads to a slight degradation due to reduced stability and exploration. Table 2 shows that methods without a reference constraint (e.g., DAPO) underperform MRPO. These results highlight the importance of the constraint as an anchor for maintaining both stability and performance.
>
> [1] Cui et al., "The Entropy Mechanism of Reinforcement Learning for Reasoning Language Models" arXiv:2505.22617.

---

> > ### Author Rebuttal · Reviewer_yLxY · 2026-03-31
> >
> > Thank you for your response. My concerns have been addressed, and I will maintain the current score.

---

> > > ### Author Response · Authors · 2026-04-04
> > >
> > > Thank you very much for reading and appreciating our rebuttal. We will revise the paper accordingly.

---

### Official Review · Reviewer_oRSC · 2026-03-13

**Soundness:** 2
**Presentation:** 2
**Significance:** 3
**Originality:** 3
**Overall Recommendation:** 4
**Confidence:** 3

**Summary:**

This work proposes an RL post training method for LLMs that roughly follows GRPO but using a (non-squared) L1 norm penalty in place of the KL regularization.
In other words, this method encourages the policy to stay close to the reference in L1 distance rather than KL divergence. A number of theoretical statements are made that characterize properties of the L1 penalty based solution and demonstrate its advantage over the KL in some cases. In addition, many experiments are carried to show superior performance compared to baselines employing other forms of regularization.

**Compliance With Llm Reviewing Policy:**

Affirmed.

**Final Justification:**

My main concerns related to the theoretical claims and their accuracy. The authors acknowledged the problems I pointed out and will amend them in their revision.
Hence my score remains unchanged.

**Key Questions For Authors:**

- Figure 1 is difficult to parse; coloring is too similar
- There is no (a) and (b) in Figure 1 (despite them being referenced from the text)
- "reward noise can induce probability fluctuations across the entire vocabulary, significantly destabilizing post-training" - Do you have examples of this? Are you referring to rewards bounded in [-1,1]? Even a toy example to illustrate the phenomenon would be great.
- It seems that recent literature on RLVR establishes SOTA to be algorithms without regularization at all (like DAPO). Then it is rather surprising that L1 penalty actually **improves** reasoning ability, while the original motivation for regularization was inaccurate rewards and potential loss of language abilities - i.e., it is not completely clear why it should help in RLVR. Do you have an explanation for why L1 improves reasoning compared to non-regularized baselines?
- The KL regularization was originally introduced in RLHF (at least in part) in order to encourage the policy to maintain language production capabilities. At the token level (where the regularization is ultimately applied), it seems important to not allow for very low probability tokens to have large probability (ie in which case they might generate non-lingual output). With this in mind, do you have a good explanation why this works so well in an RLHF setting?
- Typo: "where Gi is the set of all co-grouped tokens for the i-th token" token $\to$ group?
- The running, shortened title is still "Submission and Formatting Instructions for ICML 2026"

**Limitations:**

Not really, perhaps a short paragraph regarding limitations of the proposed method could be beneficial.

**Strengths And Weaknesses:**

### Strengths
- This work investigates an interesting replacement to standard KL based regularization and demonstrates its efficacy in a convincing manner (up to the hp-caveats mentioned below).
- Experiments are near comprehensive (up to the hp/rlhf caveats mentioned below), including two RLHF benchmarks, one with weaker reward model, two reasoning benchmarks,  and offline RL with several benchmarks, with multiple LLM models and multiple reward models.
- The cross-the-board gains in performance looks significant, if they indeed pass the bar in terms of experiment quality in hyperparameter search  and reproducibility.

### Weaknesses
- In many places, the theoretical claims are unclear, inaccurate and / or overstating; see below for a few examples.
- Given that the more conservative KL was replaced with the much less conservative (non-squared) L1-penalty, perhaps more language focused benchmark evaluations could make the submission more sound (e.g., MT-Bench / WildBench / IFEval / TruthfulQA).
- Details on hyperparameter selection in the experiments are lacking.
	- RLHF: LoRA, max-length, group-size and rollout-batch size are present, but minibatch size, learning rate, epochs, etc. are missing all or some missing in a few places. Also, was quantization used? Not clear that the provided details are enough to reproduce. Will the code be made publicly available?
	- RLVR: Same, many of the similar details seem missing.
	- It is not completely clear how hyperparameter search was conducted in each instance for MRPO variants; in some places, this is mentioned for baselines.




**Issues with statements**
- "Theorem 4.1. Consider the constrained optimization problem in Equation 1" Equation 1 is not framed as a (constrained) optimization problem (if it is constrained, it is hard to tell when it's not written as an optimization problem, without optimization variables and their domain).
- Equation 7 cannot be correct, LHS can be as large as the state space while RHS is bounded by a constant. Also not clear where $s$ on the RHS is sampled from.
- It is unclear why Theorem 5.1 implies monotonic improvement. Usually, the extra term is quadratic (while the objective linear) and as a result choosing a sufficiently small step size ensures descent (or ascent). But here the extra term is not quadratic (it is piece-wise linear). Can you add an explanation for the monotonic improvement claim? In addition, the proof is hard to follow as it (seems it) is full of typos $\pi_0, \pi_{old}, \pi_{\pi_{old}}$ - are these all the same policy?
- "Beyond stability, we analyze the efficiency of policy improvement. We quantify the improvement using the contraction coefficient" - the whole point of regularization is to not let the policy reach the un-regularized optimum. It does not really mean anything showing that one regularization is weaker (in the sense that it results in solutions closer to true optimum) than another - that might just as well be a problem, and not a feature.
- "we provide a theoretical analysis demonstrating that the L1 constraint yields a strictly superior improvement compared to KL-based methods." - Theorem 5.2 is specialized to certain scenarios, how does it establish strict superiority? In addition, it is not clear what is the formal condition used when writing e.g., $\pi_{ref} (a^*|s) \to 0$.

---

> ### Author Rebuttal · Authors · 2026-03-30
>
> ### We sincerely thank the reviewers for their insightful and constructive feedback; we are committed to addressing the technical details and clarity issues to ensure the rigor of our final manuscript.
>
> ### Weakness
> #### W1
> We thank the reviewer for pointing out these theoretical inaccuracies, we will revise them in the reversion.
>
> #### W2
> We agree that broader language-focused evaluation would strengthen the paper. Our current evaluation already includes 2 widely used language-focused benchmarks, AlpacaEval and Arena-Hard. In addition, we have obtained TruthfulQA results using EvalScope on Qwen1.5B: GRPO 24.36, MRPO-l1 37.09, MRPO-log 44.06, which are consistent with our experiments. Since these evaluation is typically computationally expensive, we will make our best effort to include more in the reversion, and others will be discussed in the limitations or discussion.
>
> #### W3
> We thank the reviewer for emphasizing reproducibility. We will add a complete configuration table including all the hyperparameters mentioned, as well as other necessary implementation details, and we will further supplement details such as the hyperparameter search procedure. We did not use quantization, and the code will be released upon publication.
>
> ### Issues with statements
> We thank the reviewer for pointing out these inaccuracies. Below we clarify these issues and indicate the corresponding revisions.
>
> #### Theorem 4.1
> We thank the reviewer for the feedback and will modify the description in the reversion by removing the term "constrained".
>
> #### Equation 7
> We agree that the current expression of Eq. 7 is imprecise and can be misleading. The intended statement is a state-conditional estimator, and we will rewrite the equation to make both the conditioning and the sampling distribution explicit.
>
> #### Theorem 5.1
> Our monotonic improvement claim is not based on a second-order Taylor argument; instead, it follows a trust-region-style performance bound, similar to the proof in TRPO. In this setting, the extra term does not need to be of higher order, because it appears explicitly in the loss. In the revision, we will add a clearer explanation of why the bound still yields monotonic improvement under the stated conditions. $\pi_0$, $\pi_{old}$, and $\pi_{\pi_{old}}$ all refer to the same policy, which we will unify in the revised version.
>
> #### Theorem 5.2
> 1) We agree that the current wording is too strong. Our intention was not to claim that being closer to the unregularized optimum is always desirable, but rather to show that L1 handles extreme scenarios more effectively, where KL-based methods may fail to achieve meaningful policy improvement ($ k \rightarrow 1 $)
> 2) The formal condition $ \pi_{\text{ref}}(a^*|s) \rightarrow 0 $ refers to scenarios where the action with the highest reward has a very low probability in the reference policy
> We apologize for the overstatement and will provide more precise theoretical statements to describe this advantage in the revised version.
>
> ### Key Questions
>
> #### Reward noise
> We refer to the phenomenon where reward noise affects all action probabilities simultaneously; for instance, given a reference policy [0.5, 0.5] and noise [0.1, -0.1], KL constraints can shift the entire distribution to [0.6, 0.4], causing the probability values of all actions to fluctuate and destabilize the training process.
>
> #### RLVR
> While RLVR relies less on regularization due to accurate rewards, the L1 penalty preserves training efficacy by preventing rapid entropy collapse and maintaining exploration, which is a well-recognized advantage of constraints [1]; existing non-regularized SOTA RLVR algorithms often incorporate new framework-level techniques for entropy maintenance, whereas our experiments focus on a pure loss-level comparison without such additional mechanisms.
>
> #### RLHF
> We agree that preserving language production capability is important. L1 can also serve this role, though differently from KL. MRPO promotes only tokens with the highest rewards; in our observations, non-lingual outputs typically do not fall into this category and are therefore suppressed, and we do not observe language degradation. Moreover, uniformly penalizing all low-probability tokens may be imprecise, since many are valid but underrepresented in pre-training data. As shown in Figure 2(b), samples for which KL is significantly larger than L1 (indicating more low-probability tokens) do not exhibit a higher proportion of reward hacking. This suggests that hacking is not concentrated in low-probability regions, thus KL’s bias against such tokens may unnecessarily restrict RL performance.
>
> #### remaining
> We sincerely thank the reviewer for the careful observation; we will correct them all in the revised version.
>
> ### Limitations
> We agree and will address them in the revised version.
>
> [1] Cui et al., "The Entropy Mechanism of Reinforcement Learning for Reasoning Language Models" arXiv:2505.22617.

---

> > ### Author Rebuttal · Reviewer_oRSC · 2026-04-03
> >
> > Thank you for your detailed reply.
> >
> > I appreciate your acknowledgement of the required amendments for the theoretical claims.
> >
> > I am still uncertain regarding Theorem 5.1 however. The argument in the original TRPO paper [1] was indeed not based on a second order approximation but it was also not rigorous (as acknowledged by the authors). They make the monotonic improvement argument for a max KL constraint (working through max L1) but then swap the max KL for average KL over the previous policy occupancy, for which they don't provide a monotonic improvement guarantee (and they also allude to this in the abstract, btw).
> >
> > > "While it is motivated by the theory, this problem is impractical to solve due to the large number of constraints. Instead, we can use a heuristic approximation"
> >
> > One can in fact obtain a monotonic improvement guarantee for TRPO/PPO but that would require a 2nd order approximation argument as I mentioned earlier.
> >
> > If you have an argument for monotonic improvement please provide a sketch for it here.
> >
> > ### Refs
> > [1] Schulman, J., Levine, S., Abbeel, P., Jordan, M. and Moritz, P., 2015, June. Trust region policy optimization. In International conference on machine learning (pp. 1889-1897). PMLR.

---

> > > ### Author Response · Authors · 2026-04-06
> > >
> > > ### Sorry for only replying to your Rebuttal Acknowledgement now. It seems I was unable to submit my comment earlier due to an issue with this system, and it has now been resolved. We thank the reviewer for the time and thoughtful feedback, and we address your remaining questions as follows.
> > >
> > > Thank you for your careful consideration of the rigor of Theorem 5.1. We agree that Theorem 5.1 is a TRPO-style argument that is not fully rigorous, and we will state this in the reversion. However, given the complexity of neural networks, deriving a fully rigorous second-order approximation argument in parameter space is extremely difficult. At present, the trust-region style proof used in TRPO remains one of the most commonly used approaches: one first establishes an upper bound on the approximation error, and then substitutes it into Eq.10 of [1] to obtain a monotonicity conclusion, even though some non-rigorous aspects do remain. Nevertheless, MRPO still goes one step further than TRPO, because MRPO proves a theoretical bound that is a function of the expectation rather than the maximum of the $ L_1 $ term, unlike the TRPO case you mentioned. Therefore, the constraint term is the same as in the actual MRPO objective, avoiding the ``heuristic approximation`` in TRPO. Hence, we believe that, at least in the example you raised, our analysis is more rigorous than TRPO and represents a further theoretical step forward. Although it is still not fully rigorous, it already reduces the gap present in TRPO. We agree that the original wording was too strong, and we will add a discussion of the non-rigorous aspects in the final version and revise the corresponding claims.
> > >
> > > Regarding the second-order approximation argument you requested, we must admit that we are currently unable to provide the most rigorous analysis in parameter space because of the complexity and unknown structure of neural networks. However, we can still provide a second-order approximation argument in policy space. Since $ L_{\pi_0}(\pi) $ and $ J(\pi) $ have the same first-order derivative with respect to $ \pi $, and $ L_{\pi_0}(\pi) $ is linear in $ \pi $, the linear terms in the Taylor expansions of $ L_{\pi_0}(\pi) $ and $ J(\pi) $, after subtracting an appropriate constant, are identical. Under bounded rewards, the extra term is a higher-order term in the norm of $ \pi - \pi_0 $. Therefore, there must exist a constant threshold such that when the policy update magnitude is smaller than this threshold (which can be controlled by adjusting $ \beta $ in MRPO), the improvement in $ L_{\pi_0}(\pi) $ is larger than the change in the extra term, and thus $ J(\pi) > J(\pi_0) $, which implies monotonic improvement. If you believe that this second-order approximation analysis in policy space is meaningful, and that it goes one step beyond TRPO from a perspective that has not been explicitly analyzed before, we can formalize it more rigorously and include it in the appendix, and also add to the discussion or limitations that we are currently unable to analyze monotonicity directly in parameter space.
> > >
> > > ### Refs
> > > [1] https://arxiv.org/pdf/1502.05477

---

### Official Review · Reviewer_v528 · 2026-03-13

**Soundness:** 3
**Presentation:** 2
**Significance:** 2
**Originality:** 3
**Overall Recommendation:** 4
**Confidence:** 2

**Summary:**

This paper proposes replacing the KL divergence constraint in RL for LLMs with an L1-norm penalty, which is motivated by two claimed limitations of KL: (1) excessive penalization of low-prior actions, and (2) sensitivity to reward noise from non-sparse updates. The L1 norm is shown to induce sparse policy updates concentrated on extreme-reward actions, with the optimal policy derived theoretically in closed form. Experiments span preference alignment (multiple 7B/8B models on AlpacaEval and Arena-Hard), RLVR (GSM8K, CodeR1), and offline RL, showcasing effectiveness of this approach across many different settings.

**Compliance With Llm Reviewing Policy:**

Affirmed.

**Final Justification:**

After reading rebuttal and other discussion, my overall assessment remains a weak accept (4). The theoretical weakness identified by reviewer oRSC regarding Theorem 5.1's monotonic improvement claim and whether it is fully rigorous seems legitimate and mostly unresolved, which prevents higher score increase. My own concerns about technical novelty are resolved by the authors' rebuttal.

**Key Questions For Authors:**

Please see weaknesses above.

**Limitations:**

Yes.

**Strengths And Weaknesses:**

## strengths
* the paper has a thorough theoretical treatment showing novel convergence guarantees and rates of improvement.
* the empirical experiments comprehensively span RLHF, RLVR, and offline settings, across multiple model families and sizes.

## weaknesses
* theory-practice gap: the theory applies to response-level one-step bandit-style MDPs, but the implementation uses token-level procedures like GRPO. In general, the theory seems to only apply for this simplistic setting
* The proof of Theorem 5.1 seems to be a straightforward adaptation of the TRPO monotonic improvement theorem (Schulman et al., 2015) with L1 in place of total variation. The lemma structure (Lemmas A.1-A.3) mirrors the original TRPO appendix closely. The paper should clarify what is technically novel in this proof and how the L1 structure specifically changes the analysis, and clarify the novelty.

---

> ### Author Rebuttal · Authors · 2026-03-30
>
> ### We thank the reviewer for the thoughtful feedback and address the weaknesses below.
>
> ### W1
> We thank the reviewer for raising this important point regarding the gap between the bandit-style analysis and token-level implementations. Our theoretical analysis indeed adopts a response-level (bandit-style) formulation for clarity. However, this formulation is standard in the analysis of RLHF-style methods, and it can naturally extend to token-level optimization by interpreting the reward $ r $ in our derivation as the action-value function Q\*
> under the optimal policy. This is because the generalized multi-step formulation is equivalent to the bandit-style optimization problem: $ \pi^* = \arg\max_{\pi} \left( \mathbb{E}_{\pi} [Q^*(s, a)] - \Omega(\pi) \right) $, where $ \Omega $ is the regularizer [1]. Importantly, our theorems are designed to study the properties of the optimal policy under L1 regularization, rather than to explicitly compute Q\* in practice. Therefore, this formulation allows the theoretical properties to carry over to the token-level updates used in GRPO-style training without compromising practical applicability. We will clarify this connection explicitly in the revision to better bridge the theory and implementation.
>
> ### W2
> We thank the reviewer for pointing out the similarity to TRPO. We agree that our proof follows a similar high-level structure to the monotonic improvement framework in TRPO, which is a standard approach for establishing policy improvement guarantees. However, proofs for this class of problems inevitably share similar high-level structures to decompose complex arguments. Nevertheless, our proof is original and differs from TRPO in both its conclusion and proof technique. In terms of the result, the objective in Theorem 5.1 is defined over the sampling distribution rather than the discounted state distribution (Eq. 1 in [2]), and it uses the expectation of the L1 norm instead of the maximum (Eq. 9 in [2]), which makes the result more practically relevant for LLM optimization. In terms of the proof technique, while TRPO relies on auxiliary event constructions  (e.g., $a'=a$), our proof is based on a recursive decomposition strategy (e.g., Lemma A.1), which follows a fundamentally different line of reasoning. The similarity in high-level structure, in our view, reflects a common pattern in this class of policy improvement proofs and is largely unavoidable. We will revise the paper to make these distinctions clearer and to better highlight the technical novelty of our analysis.
>
>
> [1] Geist et al., "A theory of regularized markov decision processes" ICML 2019.
>
> [2] Schulman et al., "Trust Region Policy Optimization" ICML 2015.

---

> > ### Author Rebuttal · Reviewer_v528 · 2026-04-03
> >
> > Thank you for the response. My concerns are addressed and I maintain my current score.

---

> > > ### Author Response · Authors · 2026-04-04
> > >
> > > Thank you very much for reading and appreciating our rebuttal. We will revise the paper accordingly.

---

### Official Review · Reviewer_YkVt · 2026-03-14

**Soundness:** 2
**Presentation:** 3
**Significance:** 3
**Originality:** 2
**Overall Recommendation:** 3
**Confidence:** 4

**Summary:**

This paper proposes to use L1 regularization to replace the KL regularization term that is widely used in RL objective for LLM alignment and reasoning. Motivations are explained on the limitations of the KL to suggest the L1 distance between the updated policy and the reference policy. Empirical performances are provided to showcase the improvement over the KL divergence baselines.

**Compliance With Llm Reviewing Policy:**

Affirmed.

**Final Justification:**

Thank the authors for their detailed response, which clarifies some of the motivations of the theorems. Yet my concerns remain about the overall novelty of the work, so I will raise my score to a 3.5 to stay neutral about the acceptance of the work.

**Key Questions For Authors:**

Q: The theorems in Section 5.2 are rather weird or claims are invalid.

In Theorem 5.1, the monotone improvement theorem is proved for L1 distance, yet: 1) L1 is TV distance for finite probability distribution, which is already proved in TRPO paper 2) the final resulting objective is clip ratio, with nothing related to the L1 regularization of the reference policy.

In Theorem 5.2, why does a smaller contraction coefficient imply better design space?

**Limitations:**

Not provided. See weakness and questions.

**Strengths And Weaknesses:**

**Strengthness:**

1. The idea of using L1 distance over the KL divergence looks new, with an unbiased estimator provided for computing the L1 distance through the rollouts.

2. Motivations are provided in using the L1 distance for LLMs.

3. The author provided some theoretical analysis in terms of convergence.

**Weakness:**

1. The overall contribution of the paper is rather incremental, which seems to be just simply replacing the KL divergence with the L1 distance, with minimal changes in other design space. There exist also a lot of existing work which has proposed to use different divergence to replace the KL divergence, yet these algorithms are not discussed or compared to.

2. The KL divergence yields the closed form of the optimal policy, yet L1 distance adds much more complexity, which is not adequately discussed.

---

> ### Author Rebuttal · Authors · 2026-03-29
>
> ### We appreciate the reviewer's careful assessment. Below we address the concerns regarding our theoretical framework and contributions to clarify the value of our work.
>
> ### W1
> We thank the reviewer for this important concern. However, our contribution is not a simple substitution, but a systematic study of how the choice of regularization fundamentally changes the optimization geometry in LLM alignment, including the geometric properties of the constraint itself, its implications for convergence behavior, and the properties of the resulting solution, as reflected in our three theorems. In this sense, our work directly addresses the critical open question of optimal regularization in LLM alignment through a systematic comparison of divergence properties, which we believe is of significant importance.
>
> This change in the regularizer affects the structure of the optimal policy, the convergence properties, and the resulting training dynamics. To support this, we provide: (1) theoretically, a comprehensive convergence analysis with entirely new proofs specifically tailored to the L1 constraint; and (2) practically, a complete L1-constrained framework consisting of four distinct algorithms across both online and offline settings, validated in diverse domains including RLVR and RLHF. Among them, the clipping in MRPO-log and the weighting schemes in the two offline algorithms are also changes in the design space.
>
> Regarding the mentioned related works, we have indeed discussed and compared them in the paper. We have already covered other constraints in the Related Work section and provided empirical comparisons with LLM-applicable baselines (e.g., DPH-F and DPH-JS) in Tables 1, 2, and 4. More broadly, the existence of these related studies itself suggests that research on alternative regularization is widely recognized as an important direction, which further supports the significance of MRPO.
>
> We will further clarify these distinctions in the revision to better highlight why MRPO is not a trivial replacement.
>
> ### W2
> The L1-constrained problem also admits a closed-form solution, which we have provided in Eq. 5. Although it involves a piecewise function, this does not necessarily affect the performance of the algorithm. If this explanation does not fully address your concern, we would appreciate further clarification on what the "complexity" specifically refers to, as our method remains computationally efficient in practice.
>
> ### Q1
> 1) If one considers the terminology alone, L1 is indeed closely related to TV distance. However, in RL, the form of the constraint must be understood together with where it is imposed and under which state distribution it is evaluated. Under this perspective, the constraint in MRPO is fundamentally different: (i) TRPO utilizes the maximum TV divergence across all states, whereas our proof is based on the expected value; and (ii) the state distribution in MRPO, as presented in our proof, is the sampling distribution, whereas in TRPO it takes a more complex weighted-sum form. These differences make MRPO much easier to implement in practice than TRPO.
> 2) The MRPO objective does explicitly include the L1 regularization term (Eqs. 8–11). The clipping mechanism is a standard variance-reduction technique widely used in algorithms such as PPO and GRPO to prevent instability caused by extreme samples. In practice, the clipping trigger rate is extremely low and does not undermine the overall convergence properties. This is also one of the reasons why many existing works did not perform additional convergence analysis.
>
> ### Q2
> As defined in the preamble to Theorem 5.2, a smaller contraction coefficient directly implies a tighter bound to the optimal policy, thereby leading to superior training outcomes. If this explanation does not fully address your concern, we would appreciate a more detailed clarification of your specific interest in the "design space".

---

> > ### Author Rebuttal · Reviewer_YkVt · 2026-04-02
> >
> > Thank you for the authors' responses. After reading, my concerns are still not addressed:
> >
> > Q1: The clipping rate in PPO is in the same spirit of adding KL regularization in TRPO to prevent aggressive update, NOT variance reduction. I also checked the proof of the paper, there is no fundemental difference in the proof techniques compared to TRPO.
> >
> > Q2: My concern on the "design space" is simply confusion on the authors' claims. Why a tighter bound to the optimal policy imply the advantage of L1 regularization over KL? This claim is not convincing.

---

> > > ### Author Response · Authors · 2026-04-04
> > >
> > > ### We thank the reviewer for the careful reading and feedback. We address the remaining questions below.
> > >
> > > ### Q1
> > > We agree with the reviewer on the relationship between PPO clipping and KL regularization in theory. However, beyond clipping, PPO also allow an explicit KL regularization term, and this explicit regularizer is widely used in RL for LLMs; such an explicit KL term appears in Eq. 2 of [1], in mainstream training frameworks such as [2], and in later PPO-style variants such as Eq. 3 of [3]. MRPO discusses this explicit regularization term, while retaining the clipping term in practice. Because clipping is retained, we acknowledge that Theorem 5.1 is not a strictly rigorous direct convergence proof for the MRPO objective, but rather shows that replacing KL by L1 still admits a TRPO-style monotonic-improvement control, and we will state this limitation in the discussion and revise the corresponding wording in the paper.
> > >
> > > We agree that, at a high level, the convergence proof of Theorem 5.1 may appear similar in spirit to that of TRPO, since both are based on deriving an upper bound on the error term. However, we still believe Theorem 5.1 is necessary. Unlike TRPO, the bound required by MRPO is the expected L1 term under the sampling policy distribution ($ E_{p(s_t|\pi_0)} $) in Theorem 5.1, whereas the TRPO result is stated in terms of the maximum TV divergence ($ \max_{s} D_{TV} $) in Eq. 7 of [4], and TRPO explicitly notes that it does not prove the relation to the expectation form ("Instead, we can use a heuristic approximation which considers the average"). These are different, so the original TRPO result cannot help MRPO directly, and MRPO must introduce Theorem 5.1 for this purpose. To prove this different bound, Theorem 5.1 also uses a proof strategy different from TRPO: (1) TRPO first proves that the bound can be characterized by the statistic of an $ \alpha $-coupled policy pair (Eq. 45 in [4]), and then connects it to the maximum TV divergence (at the end of Lemma 3), whereas MRPO directly proves that the expected L1 term itself forms the bound; (2) TRPO is based on the notion of an $ \alpha $-coupled policy pair (Definition 1 in [4]), while the proof of MRPO is based on the recursive property of the Markov decision process (our Lemma A.1), which is a different proof route. Therefore, we view Theorem 5.1 as a generalization of the TRPO-style theorem to our setting. Although it can indeed be regarded as a follow-up type of result, it is still necessary for the theoretical analysis of MRPO, and we will clarify the role and scope of Theorem 5.1 more carefully in the revised version.
> > >
> > > ### Q2
> > > In some extreme cases, for example when $ \pi_{ref}(a^*) \rightarrow 0 $, the distance to the optimal policy before and after training under the KL constraint remains nearly the same ($ k \rightarrow 1 $); although it is decreasing, the amount of decrease tends to 0, while L1 does not have this issue. In such cases, regardless of how $ \beta $ is chosen, MRPO stays closer to the optimal policy, which means that in this regime the L1 trust region always allows a more substantial update toward the optimal policy. This is what we mean by "advantage" here.
> > >
> > > As we also explained in our response to Reviewer oRSC under "Issues with statements", we agree that Theorem 5.2 is somewhat overstated in its current form. In fact, it does not claim that L1 regularization is always better than KL regularization; it only establishes this in this special case. However, we believe this special case is very common in LLM settings, because $ \pi_{ref}(a^*) \rightarrow 0 $ means that the optimal response has near-zero probability under the base model, which is common in practice; for example, on some problems, an LLM may fail to answer correctly even after thousands of attempts. Therefore, MRPO has a stronger advantage over KL regularization in LLM settings of this kind, and we will add this clarification in the discussion.
> > >
> > >
> > > [1] https://arxiv.org/pdf/1909.08593
> > >
> > > [2] https://github.com/huggingface/trl/blob/v0.25.0/trl/trainer/ppo_trainer.py#L598
> > >
> > > [3] https://arxiv.org/pdf/2402.03300
> > >
> > > [4] https://arxiv.org/pdf/1502.05477

---

### Decision · Program_Chairs · 2026-04-30

**Decision:**

Accept (regular)

**Comment:**

**Summary:** This paper proposes to use L1 regularization to replace the KL regularization term that is widely used in the RL objective for LLM alignment and reasoning. The L1 norm is shown to induce sparse policy updates concentrated on extreme-reward actions, with the optimal policy derived theoretically in closed form. Experiments span preference alignment (multiple 7B/8B models on AlpacaEval and Arena-Hard), RLVR (GSM8K, CodeR1), and offline RL, showcasing the effectiveness of this approach across many different settings.

**Strengths:**
- Well-written paper that motivates the use of L1 regularization
- Extensive empirical evaluation showing that the use of L1 regularization can improve the performance across a variety of settings

**Weaknesses:**
- Some of the theoretical claims are overstating the result
- Details on hyperparameter selection in the experiments is lacking

**Decision and Suggested Changes:** This paper was on the borderline. After carefully reading the rebuttal/discussion, I went through the paper myself. I think the paper's contributions merit acceptance. Addressing the following concerns will help strengthen the current version of the paper:
- Clearer explanation for the use of clipping (ensuring that the deviation from the current policy is controlled) vs that of the L1 constraint (ensuring proximity to the reference policy), and toning down the claim about the theoretical derivation of the final MRPO objective (Rev. YkVt)
- Improve the writing of the proof of Theorem 5.1, and explicitly clarify the difference with the TRPO derivation (Rev. YkVt, Rev. v528, Rev. oRSC)
- Include the configurations for all hyperparameters and additional implementation details, with additional language-focused evaluation (Rev. oRSC)
- Include toy example to illustrate that reward noise can affect all action probabilities simultaneously (Rev. oRSC).
- Evidence that using MRPO does not result in an entropy collapse (Rev. yLxY)